# Quantum annealing of a frustrated magnet

Yuqian Zhao[1], Zhaohua Ma[1], Zhangzhen He [2], Haijun Liao[3,4], Yan-Cheng Wang[5,6] ✉, Junfeng Wang [1] & Yuesheng Li [1] ✉

Quantum annealing, which involves quantum tunnelling among possible solutions, has state-of-the-art applications not only in quickly finding the lowest-energy configuration of a complex system, but also in quantum computing. Here we report a single-crystal study of the frustrated magnet $\alpha$-$CoV_2O_6$, consisting of a triangular arrangement of ferromagnetic Ising spin chains without evident structural disorder. We observe quantum annealing phenomena resulting from time-reversal symmetry breaking in a tiny transverse field. Below ~ 1 K, the system exhibits no indication of approaching the lowest-energy state for at least 15 hours in zero transverse field, but quickly converges towards that configuration with a nearly temperature-independent relaxation time of ~ 10 seconds in a transverse field of ~ 3.5 mK. Our many-body simulations show qualitative agreement with the experimental results, and suggest that a tiny transverse field can profoundly enhance quantum spin fluctuations, triggering rapid quantum annealing process from topological metastable Kosterlitz-Thouless phases, at low temperatures.

Finding the best solution to complex problems is a central challenge in many fields, spanning from mathematics and computer science to statistical physics. The prototypical problem for studying lowest-energy configurations in many-body systems is likely probing the ground state of a correlated Ising model[1–7]. The annealing of a complex Ising spin system towards its optimal state can be time-consuming, and typically characterized by a relaxation time constant $\tau$[8]. For a thermal (or classical) annealing, $\tau$ rapidly approaches infinity as temperature ($T$) decreases to 0 K, following a thermally activated Arrhenius form, $\tau = \tau_m \exp(\Delta E / T)$, where $\tau_m$ and $\Delta E$ represent the relaxation time at high temperatures and the barrier energy, respectively. At low temperatures, this classical form hinders the system from converging towards the lowest-energy configuration, resulting in reduced work efficiency. In contrast, quantum annealing (QA) with a transverse field exhibits superiority over classical annealing[3,9], resulting in a much shorter $\tau$ that is temperature-independent as $T \to 0$ K.

Various Ising models have recently been simulated using programmable superconducting QA processors[4], Rydberg atoms[10], and other approaches, yielding exotic many-body phenomena such as dynamical phase transitions and topological configurations[5,6,11]. While these simulations are of great interest as quantum simulators, ultimately, research efforts should be directed towards exploring and developing real-world materials. However, real materials are typically highly complex systems that are difficult to model due to the numerous perturbation interactions arising from structural imperfections[12–14]. Thus, experimentalists and materials scientists have been making significant efforts to search for "ultra-clean" materials that exhibit precisely solvable models, allowing for the observation of well-defined many-body phenomena[15]. To the best of our knowledge, there exists only one reported example of an Ising spin glass, $LiHo_xY_{1−x}F_4$, that exhibits many-body QA phenomena[9,16,17]. However, the site-mixing disorder of Ho and Y introduces interaction randomness, making it challenging to create a precise microscopic model of this system. Moreover, in this compound, QA phenomena are only visible at very low temperatures, much lower than $\Delta E \leq 0.54$ K[16], due to the weak couplings between the rare-earth magnetic moments. The triangular-lattice Ising antiferromagnet $TmMgGaO_4$ also exhibits interaction randomness[14], but no QA has been reported for this

[1]Wuhan National High Magnetic Field Center and School of Physics, Huazhong University of Science and Technology, 430074 Wuhan, China. [2]State Key Laboratory of Structural Chemistry, Fujian Institute of Research on the Structure of Matter, Chinese Academy of Sciences, 350002 Fuzhou, China. [3]Institute of Physics, Chinese Academy of Sciences, P.O. Box 603, 100190 Beijing, China. [4]Songshan Lake Materials Laboratory, 523808 Dongguan, China. [5]Hangzhou International Innovation Institute, Beihang University, 311115 Hangzhou, China. [6]Tianmushan Laboratory, 311115 Hangzhou, China. ✉e-mail: ycwangphys@buaa.edu.cn; yuesheng_li@hust.edu.cn

material. The frustrated spin-chain compound $Ca_3Co_2O_6$ has no apparent structural disorder and was predicted to exhibit QA using a D-WAVE QA computer[18], but no measurable effect of QA has been observed in this material[19]. Therefore, further investigation and exploration of other candidate materials for realizing QA is required.

Previous studies on $\alpha$-$CoV_2O_6$ have suggested that the compound can experimentally realize the spatially anisotropic triangular lattice of ferromagnetic Ising spin chains (Fig. 1a) with no apparent structural disorder[20,21]. Additionally, high-quality single crystals of this compound are available for further investigation[22]. However, no reports on quantum effects of transverse magnetic fields have been made in $\alpha$-$CoV_2O_6$. Here, we propose the use of such an "ultra-clean" magnetic material for QA studies, utilizing ultra-low-$T$ measurements of physical properties and large-scale Monte Carlo (MC) computations, in transverse magnetic fields. Under zero applied transverse field and at temperatures below ~ 2 K, the frustrated spin system has a strong tendency to get stuck in metastable Kosterlitz-Thouless (KT) phases characterized by the appearance of topological vortices and antivortices[23,24] around the domain walls. By contrast, a small transverse field, achieved by breaking the time-reversal symmetry in an applied transverse magnetic field, can profoundly enhance quantum-mechanical tunneling at low temperatures, triggering QA towards the optimum state with a short and nearly temperature-independent relaxation time.

## Results

### Spin Hamiltonian

In $\alpha$-$CoV_2O_6$, the 28 electronic states of $Co^{2+}$ ($^4F$) linearly superpose into 14 doublets under the crystal electric field and spin-orbit coupling, preserving the time-reversal symmetry as described by the Kramers theorem. The lowest-lying doublet ($|E_1\rangle$ and $|E_2\rangle$) is well-separated from the first excited doublet with an energy gap of $E_3 - E_1$ ~ 140 K, indicating the effective spin-1/2 dipole moment of $Co^{2+}$ with Ising anisotropy ($g$ factors of $g^x$ ~ $g^y$ ~ 0 and $g^z$ ~ 11, see Supplementary Note 1 and Supplementary Fig. 1) at low temperatures. Under nonzero transverse magnetic field, the time-reversal symmetry is broken, which results in the slight splitting of the ground-state doublet with an inner gap of $\Gamma = E_2 - E_1$ (Fig. 1b, c). Hence, transverse-field terms,

$\mathcal{H}_{TF} = -\Gamma \sum_i S_i^{x}$[14,25], are induced in the Ising spin Hamiltonian of $\alpha$-$CoV_2O_6$ (Supplementary Note 2 and Supplementary Fig. 3), which no longer commutes with $S_i^z$. The transverse-field terms cause the quantum tunneling between states with $S_i^z = \pm 1/2$, compete with the Ising couplings (Fig. 1a), and thus suppress the antiferromagnetic ordering temperature of $T_N$ (Fig. 1c). In a linear relationship between $T_N$ and $\Gamma$ as $\Gamma$ approaches 0 K[26], the single-ion Hamiltonian shows good agreement with experimental results, as depicted in Fig. 1c.

The exchange couplings beyond fourth-nearest neighbors are neglected in $\alpha$-$CoV_2O_6$, based on the previously reported density functional theory calculation[21]. The original three-dimensional (3D) Hamiltonian is formulated with the transverse field per spin,

$$\mathcal{H} = J_0 \sum_{\langle i,i_0 \rangle} S_i^z S_{i_0}^z + J_1 \sum_{\langle i,i_1 \rangle} S_i^z S_{i_1}^z + J_2 \sum_{\langle i,i_2 \rangle} S_i^z S_{i_2}^z + J_3 \sum_{\langle i,i_3 \rangle} S_i^z S_{i_3}^z$$
$$- \mu_0 \mu_B H^z g^z \sum_i S_i^z - \Gamma \sum_i S_i^x. \tag{1}$$

After fitting the quasi-equilibrium-state thermodynamic data measured above 1.9 K at $\Gamma = 0$ K, we have refined the strengths of the couplings (Supplementary Note 3). As a result, we obtained $J_0 = -30.73$ K, $J_1 = 3.60$ K, $J_2 = 14.21$ K, and $J_3 = 2.55$ K, with an improved goodness of fit (Supplementary Table 1, Supplementary Figs. 5, 6).

The original 3D Hamiltonian with $\Gamma$ ~ 0 K (Eq. (1)) analytically yields three different lowest-energy phases separated by two critical longitudinal fields, $\mu_0 H_{c1}^z = \frac{2J_1+J_2-4J_3}{\mu_B g^z}$ (~ 1.5 T) and $\mu_0 H_{c2}^z = \frac{2J_1+J_2+2J_3}{\mu_B g^z}$ (~ 3.6 T), as shown in Fig. 1d. At $|H^z| < H_{c1}^z$, the ground phase is a stripe antiferromagnetic state with zero longitudinal magnetization $M^z = 0$ $\mu_B$/Co, at $H_{c1}^z < |H^z| < H_{c2}^z$ the system enters a "up-up-down" state with $|M^z| = g^z/6$ ~ 1.8 $\mu_B$/Co, whereas at $|H^z| > H_{c2}^z$ all the Ising spins are fully polarized with $|M^z| = g^z/2$ ~ 5.5 $\mu_B$/Co. These lowest-energy configurations have been verified by neutron diffraction measurements[20].

### Superiority of quantum annealing over thermal annealing

All three lowest-energy configurations of $\alpha$-$CoV_2O_6$ have very different longitudinal magnetization values, and thus we measured the dc magnetization to assess the convergence of the annealing process. For comparison, at each low temperature the system was initialized with

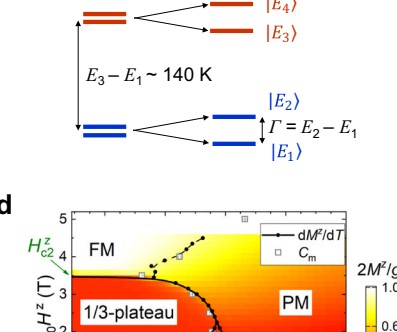

**Fig. 1 | Frustrated Ising model of $\alpha$-$CoV_2O_6$. a** The crystal structure. The ferromagnetic intrachain interaction ($J_0$) and antiferromagnetic interchain interactions ($J_1, J_2, J_3$) of the original Hamiltonian are indicated. The unit cell is depicted with thin lines and the coordinate system for the spin components is established. **b** Simplified schematic of the four lowest-lying crystal-electric-field levels of the $Co^{2+}$ ($^4F$) ion. The ground-state Kramers doublet at zero applied transverse magnetic field ($H^\perp = 0$) is lifted and a small gap of $\Gamma = E_2 - E_1$ is opened due to the time-reversal symmetry breaking in $H^\perp \neq 0$. **c** The decrease in the Néel transition

temperature at $H^\perp$, $T_N(0\,T) - T_N(\mu_0 H^\perp)$, measured by specific heat (Supplementary Fig. 2). Error bars, $1\sigma$ s.e., and the colored lines present the $H^\perp$ dependence of $\Gamma$ calculated from the single-ion Hamiltonian (see Supplementary Note 2). **d** Quasi-equilibrium-state temperature-longitudinal field ($\mu_0 H^z$) phase diagram of $\alpha$-$CoV_2O_6$ (Supplementary Fig. 4). Four phases are indicated: Stripe antiferromagnetic (AF), 1/3-plateau magnetization, fully ferromagnetic (FM), and paramagnetic (PM) phases, along with two critical longitudinal fields.

the same fully-inversely-polarized state in $\mu_0 H^z = -4.2$ T, and then the external longitudinal $\mu_0 H^z$ was quickly raised to 2 T with a constant ramp rate $\mu_0 \mathrm{d}H^z/\mathrm{d}t = 10$ mT/s. As $\mu_0 H^z$ reaches 2 T, we started measuring the time dependence of $M^z$.

In zero transverse magnetic field $H^x \sim 0$, as shown in Fig. 2a, for instance, the longitudinal magnetization keeps at low values ($< 0.2 \mu_B/$Co), without showing any tendency towards the one-third plateau value of $\sim g^z/6$ at least up to 15 hours, indicating that the system gets deeply stuck in metastable states, below $\sim 1$ K. We also performed the similar experiments with $H^x \sim 0$ at various temperatures. Above $\sim 1$ K, $M^z$ slowly relaxes to a value that monotonically decreases from $\sim 1.7 \mu_B/$Co (at 2.8 K) to 0.3 $\mu_B/$Co (at 1.3 K), within our 5-hour experimental time window. In sharp contrast, when a transverse magnetic field is applied, $H^x \sim H^z$, $M^z$ rapidly relaxes to $\sim 1.4 \mu_B/$Co with a time constant of $\tau \sim 10$ s as shown in Fig. 2c, suggesting successful annealing through enhanced quantum fluctuations (Fig. 2e). Moreover, we repeated the experiments with $H^x \sim H^z$ at various temperatures down to 80 mK, which confirm that the system quickly anneals to high-magnetization states with long-time $M^z$ ranging from $\sim 1.4 \mu_B/$Co (at low temperatures) to $\sim 1.8 \mu_B/$Co (at high temperatures), as soon as $\mu_0 H^z$ reaches 2 T from $-4.2$ T. Alternately, one can anneal the system thermally by increasing the temperature (Fig. 2d). When the temperature is raised to 1.85 K from 0.5 K, $M^z$ relaxes to a high value of $\sim 1.2 \mu_B/$Co (Fig. 2b), following a stretching-exponent behavior[27], with a time constant of $\tau = 2{,}550(30)$ s much longer than the quantum counterpart. The measured stretching exponents $\beta'$ range from $\sim 0.5$ to $\sim 0.9$ at $T \leq 2$ K (refer to the inset of Supplementary Fig. 14d), which are roughly consistent with the MC calculations. The relaxation processes of independent samples have

varying durations, resulting in a substantial distribution of relaxation times and thus exhibiting a pronounced stretched-exponential relaxation behavior[8].

We also measured the isothermal magnetic hysteresis loops at 0.5 and 1.8 K as shown in Fig. 2f, h, respectively. At longitudinal magnetic fields $\mu_0|H^z| > \mu_0 H^z_{c1} \sim 1.5$ T, the transverse magnetic field clearly suppresses the hysteresis loop (Fig. 2g, i). In the stripe antiferromagnetic phase ($|H^z| < H^z_{c1}$), the internal transverse magnetic fields contributed by the ordered dipole moments of Co are calculated as $\mu_0 H^x_{in} \sim \pm 0.2$ T on the spin-up and -down sites, respectively. The observation of narrowed loop widths in zero external transverse field at $|H^z| < H^z_{c1}$ may be attributed to these internal dipole fields.

At a low temperature of 1.8 K, the reduce of annealing time $\tau$ by transverse fields is most profound under the longitudinal field of $\mu_0 H^z = 2$ T: $\tau = 3{,}378$ s at $H^x \sim 0$ is progressively suppressed to 89 s at $H^x \sim H^z$ and 66 s at $H^x \sim 1.7 H^z$, respectively (Fig. 3a). At this longitudinal-field strength, slightly above the critical field $\mu_0 H^z_{c1} \sim 1.5$ T, the Zeeman interaction drives the frustrated spin system close to the critical point. This configuration results in the highest concentrations of domain walls and topological defects along with the longest relaxation time at $\mu_0 H^z = 2$ T, according to MC simulations (Supplementary Figs. 7, 8). Consequently, QA is most clearly seen at $\mu_0 H^z = 2$ T when a transverse field is applied. We focused on the relaxation measurements on $\alpha$-CoV$_2$O$_6$ at $\mu_0 H^z = 2$ T. At $H^x \sim 0$, the thermally activated Arrhenius behavior of $\tau$ is clearly observed down to the low temperatures (where $\tau \to \infty$, see Fig. 3b), $\tau^{-1} = \tau^{-1}_m \exp(-\Delta E/T)$. Here, $\tau^{-1}_m = 1.5 \pm 0.1$ Hz, and the fitted barrier energy $\Delta E(\mu_0 H^z = 2$ T$) = 15.3 \pm 0.2$ K, comparable with the strengths of the interchain couplings of $\alpha$-CoV$_2$O$_6$, is significantly

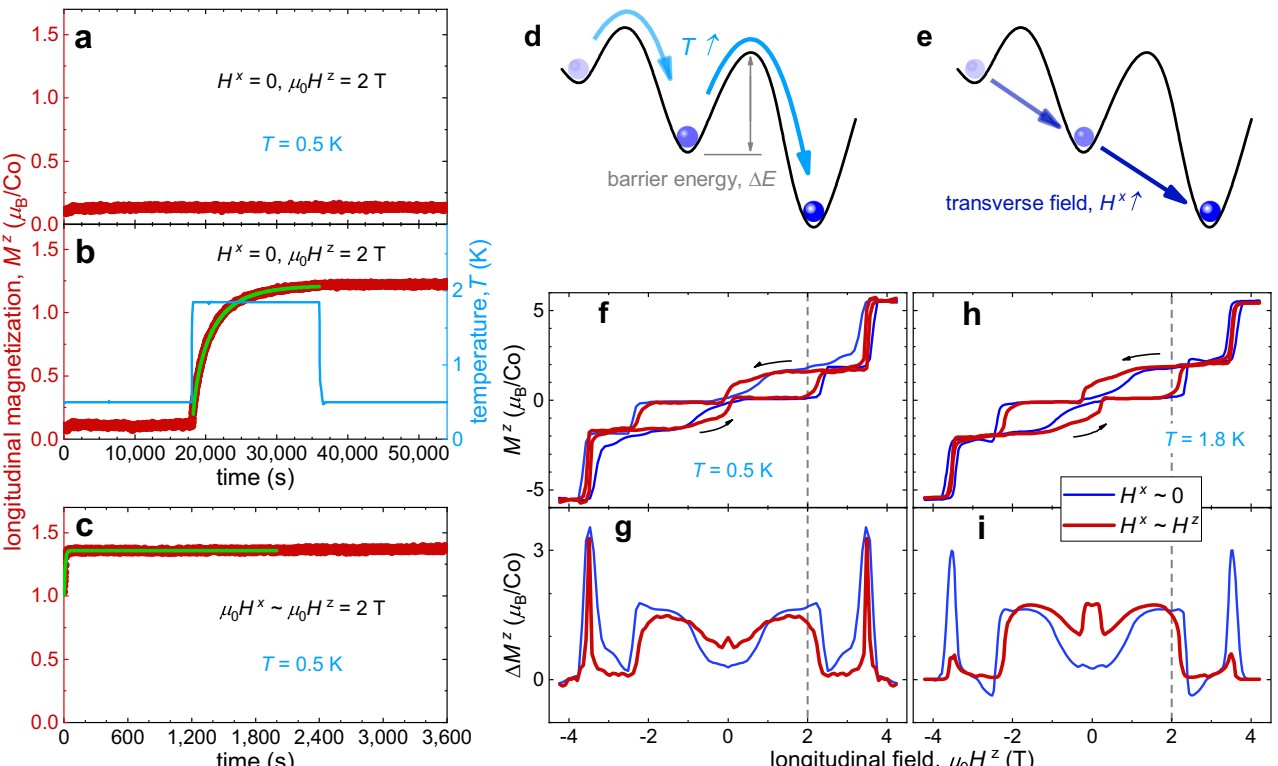

**Fig. 2 | Quantum annealing versus thermal annealing. a–c** The temporal variation of magnetization measured on $\alpha$-CoV$_2$O$_6$. We define $t = 0$ as the moment when the field component along the $z$-axis ($\mu_0 H^z$) reaches 2 T from $-4.2$ T (where all spins are fully-inversely-polarized), at a temperature of 0.5 K and with a constant ramp rate of $\mu_0 \mathrm{d}H^z/\mathrm{d}t = 10$ mT/s. As shown in (**a**), under zero transverse field the system is frozen into metastable states with small longitudinal magnetization of $M^z \sim 0.1 \mu_B/$Co at low temperatures. By raising the temperature $T$ or applying a transverse magnetic field $H^x$, the system can relax towards the equilibrium state

with $M^z = g^z/6 \sim 1.8 \mu_B/$Co through thermal or quantum annealing, respectively. The green lines in (**b**) and (**c**) represent the stretched-exponential fits, $M^z(t') = (M_0 - M_\infty) \exp[-(t'/\tau)^{\beta'}] + M_\infty$, where $\tau$ is the relaxation time, $\beta'$ is the stretching exponent, $M_0$ and $M_\infty$ are the initial and final magnetization, and $t' = t - t_0$ with $t_0$ presenting the start time of the fit. **d, e** Sketches of thermal (by increasing $T$) and quantum (by applying $H^x$) annealing. **f, h** $M^z$-$H^z$ hysteresis loops measured at 0.5 and 1.8 K with $H^x \sim 0$, $H^z$. **g, i** The loop widths of the curves in (**f, h**).

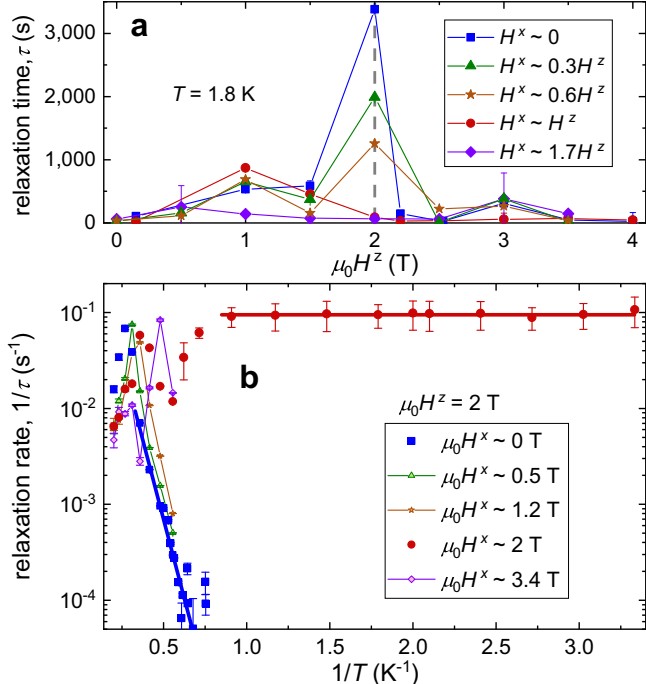

**Fig. 3 | Annealing time constants. a** The relaxation time $\tau$ of $\alpha$-$CoV_2O_6$ measured with different applied transverse fields ($H^x$) with respective to the longitudinal field ($H^z$), $H^x \sim 0$, $0.3H^z$, $0.6H^z$, $H^z$, $1.7H^z$, respectively. **b** Inverse-temperature dependence of the relaxation rate $\tau^{-1}$. The blue line represents the fit to the experimental data with $H^x \sim 0$ by the Arrhenius function $\tau^{-1} = \tau_m^{-1}\exp(-\Delta E/T)$ below 2.8 K, whereas the red line shows a constant fit to the data with $\mu_0 H^x \sim 2$ T below 1.1 K. The raw data of magnetization relaxations, used in extracting $\tau$, are shown in Supplementary Fig. 14. Error bars, $1\sigma$ s.e.

larger than the barrier energy reported in LiHo$_x$Y$_{1−x}$F$_4$ ($\le 0.54$ K)[16]. In sharp contrast, the relaxation rates measured at the transverse magnetic field of $\mu_0 H^x \sim 2$ T are nearly temperature-independent at low temperatures, $\tau^{-1} \sim 0.1$ Hz, suggesting a clear QA effect.

By conducting systematic measurements of magnetic hysteresis loops and relaxations, we observed a distinct classical-quantum crossover behavior induced by the transverse magnetic field below $\sim 4$ K. This behavior is illustrated in Fig. 3, and further details can be found in Supplementary Note 6 and Supplementary Fig. 17. The temperature-dependent relaxation rate peaks at $T \sim 3$-4 K ($1/T \sim 0.27$-$0.34$ K$^{-1}$, see Fig. 3b), indicating a maximum in spin fluctuations. This is likely due to a crossover in spin dynamics driven by thermal fluctuations with rising temperature. Notably, this observed crossover is reminiscent of similar findings previously reported in LiHo$_x$Y$_{1−x}$F$_4$[16].

**Quantum Monte Carlo simulations**

The above observations demonstrate the occurrence of QA in $\alpha$-$CoV_2O_6$ and provide experimental evidence supporting its superiority over thermal annealing, as previously proposed in theoretical work on the Ising spin glass[3]. Although we have presented the processes of quantum and thermal annealing towards the lowest-energy configuration in experiments, it remains unclear whether the microscopic model of Eq. (1) can capture the observed QA effect of $\Gamma$. To investigate this, we have conducted stochastic series expansion (SSE) quantum Monte Carlo (QMC) simulations at low temperatures[28,29]. The microscopic model (Eq. (1)) has shown good agreement with both the quasi-equilibrium-state and dynamic properties of $\alpha$-$CoV_2O_6$ at $\Gamma = 0$ K (see Supplementary Figs. 8, 10). We further use it to study the quasi-equilibrium and out-of-equilibrium properties of the system at $\Gamma > 0$ K.

Compared with the quantum Ising spin-glass systems[30], in $\alpha$-$CoV_2O_6$ achieving critical suppression of the ground-state magnetic

ordering necessitates high transverse fields of $\Gamma_c \sim 36$ and 28 K (comparable to $|J_0|$) at $\mu_0 H^z = 0$ and 2 T, respectively, as estimated using the SSE method based on the original 3D Hamiltonian (Fig. 4a, b). This would require excessively high transverse magnetic fields of $> 50$ T, leading to significant stress within the single crystal and potential sample destruction due to the strong Ising magnetic anisotropy[31]. The primary focus of this work is to investigate the QA effect in a tiny transverse field, $\Gamma \ll \Gamma_c$.

The transverse field is expected to induce quantum-mechanical tunneling between states with $S_i^z = \pm 1/2$, thereby causing QA. To quantify this more precisely, we estimate the quantum effects of the tiny transverse field using a single-site Hamiltonian in the molecular-field approximation, $\mathcal{H}_s = -hS^z - \Gamma S^x$, where $h \sim |J_0|$ represents the mean field. Through exact diagonalization of $\mathcal{H}_s$, we calculate the density elements, $W_\uparrow = \langle\uparrow|\exp(-\beta\mathcal{H}_s)|\uparrow\rangle$, $W_\downarrow = \langle\downarrow|\exp(-\beta\mathcal{H}_s)|\downarrow\rangle$, and the matrix ratios $W_\downarrow/W_\uparrow$ as shown in Fig. 4c, where $\beta \equiv 1/(k_B T)$. When applying a small transverse field ($\Gamma \le 0.1$ K), the probability of flipping the spin, as indicated by $W_\downarrow/W_\uparrow$, can be significantly enhanced at $T \le 1.8$ K. Therefore, $\Gamma$ facilitates quantum-mechanical tunneling between different states, likely inducing the QA effects observed at low temperatures in $\alpha$-$CoV_2O_6$. It is noteworthy that the simplified molecular-field model cannot directly account for the observed very slow spin dynamics and, consequently, the many-body QA effects. To investigate these effects more precisely, we employ SSE-QMC simulations for small transverse fields using the original Hamiltonian of $\alpha$-$CoV_2O_6$, Eq. (1) (Fig. 4d–i).

When $\Gamma$ raises to 0.1 K, for example by applying $\mu_0 H^y \sim 10$ T on $\alpha$-$CoV_2O_6$, the energy (per site) is significantly reduced (see Fig. 4d, e) and the longitudinal magnetization at $\mu_0 H^z = 2$ T is profoundly increased (see Fig. 4f) after the same large MC steps (MCS). The equilibrium-state energy and magnetization decreases induced by $\Gamma$, estimated using $\mathcal{H}_s$, are entirely negligible, $|\Delta E_\Gamma| \sim \sqrt{\Gamma^2 + h^2}/2 - h/2 \le 8\times10^{-5}$ K and $|\Delta M_\Gamma^z| \le 3\times10^{-5}\mu_B/$Co, at $\Gamma \le 0.1$ K. Clearly, the observed energy decrease of $> 0.2$ K caused by $\Gamma$ ($\le 0.1$ K) at large MCS (see Fig. 4d, e) cannot be attributed to the above equilibrium-state energy decrease. The observed increase in the longitudinal magnetization caused by $\Gamma$ ($\le 0.1$ K) at large MCS (see Fig. 4f) also contradicts the equilibrium-state magnetization decrease. Furthermore, the energies calculated at various transverse fields ($\Gamma \le 0.1$ K) align with each other at small MCS, and the distinct energy decrease caused by $\Gamma$ becomes apparent only at larger MCS and lower temperatures, in the QMC simulations (Fig. 4d, e). Therefore, these simulations for both energies and longitudinal magnetization unequivocally demonstrate QA effects caused by the small transverse field ($0.02 \le \Gamma \le 0.1$ K, $\ll \Gamma_c$) at low temperatures ($\sim 1$ K), providing a qualitative description of the previously mentioned experimental observations. Moreover, our heat transport measurements, which also exhibit clear QA effects around 1 K, were conducted at a transverse field of up to 0.07 K (Fig. 5), significantly smaller than $\Gamma_c \sim 28$ K, and more than three times larger than 0.02 K.

At $T = 1$ K, the ratios $W_\downarrow/W_\uparrow$ at $\Gamma = 3.5$ mK and 0.02 K are estimated to be $\sim 3\times10^{-9}$ and $1\times10^{-7}$, respectively, markedly larger than $W_\downarrow/W_\uparrow \sim 5\times10^{-14}$ at $\Gamma = 0$ K (Fig. 4c). To observe clear QA effects of $\Gamma$, a minimal number of MCS roughly proportional to $\sim W_\uparrow/W_\downarrow$ is required, resulting in $W_\uparrow/W_\downarrow \sim 3\times10^8$ ($\Gamma = 3.5$ mK) and $9\times10^6$ ($\Gamma = 0.02$ K). Although $W_\downarrow/W_\uparrow$ is also significantly enhanced by $\Gamma = 3.5$ mK, much larger MCS are needed to observe clear QA effects compared to $\Gamma = 0.02$ K. At $\Gamma = 0.02$ K, QA effects are evident only after $\sim 3\times10^4$ SSE MCS ($\times100$) (Fig. 4e, f), necessitating more than $\sim 1\times10^6$ SSE MCS ($\times100$) at $\Gamma = 3.5$ mK. Therefore, observing clear QA effects at $0 < \Gamma < 0.02$ K remains extremely challenging due to the increased computational cost associated with larger MCS.

While the present microscopic model of Eq. (1) is precise enough to account for the quasi-equilibrium-state thermodynamic properties (Supplementary Fig. 10), it may lack high precision in simulating the slow spin dynamics observed in $\alpha$-$CoV_2O_6$, especially at smaller

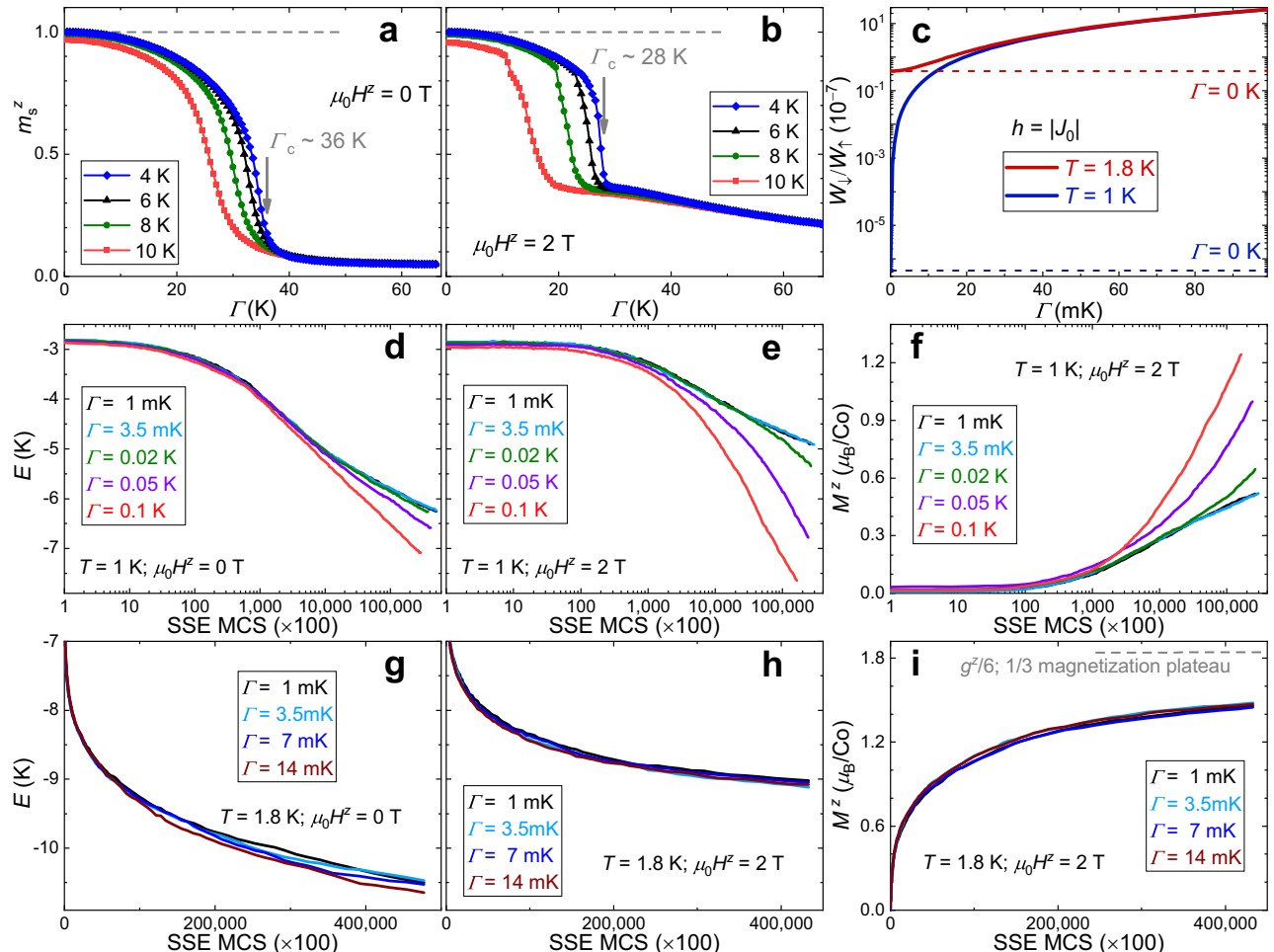

**Fig. 4 | Simulations of annealing processes. a, b** Transverse field dependence of staggered magnetization $m_s^z = \langle|\sum_j S_j^z(-1)^{j_a+j_c}|\rangle/(NS)$ and $3\langle|\sum_j S_j^z \exp[i2\pi(j_a + j_c)/3]|\rangle/(2NS)$ calculated at $\mu_0H^z = 0$ and 2 T, respectively. Here, $N$ is the site number, $S = 1/2$, and $j_a$ and $j_c$ are site indexes along the $a$- and $c$-axis, respectively. The critical transverse fields $\Gamma_c$ are marked. **c** Density matrix ratios, $W_\downarrow/W_\uparrow$. Here, $W_\uparrow = \langle\uparrow | \exp(-\beta\mathcal{H}_s) | \uparrow\rangle$ and $W_\downarrow = \langle\downarrow | \exp(-\beta\mathcal{H}_s) | \downarrow\rangle$ are exactly calculated using the single-site molecular-field Hamiltonian $\mathcal{H}_s = -hS^z - \Gamma S^x$. **d, e, g, h** Monte Carlo step (MCS) dependence of the energy per site ($E$) calculated at $T = 1$, 1.8 K in $\mu_0H^z = 0$, 2 T.

**f, i** MCS dependence of longitudinal magnetization calculated at $\mu_0H^z = 2$ T. The original spin Hamiltonian of $\alpha$-CoV$_2$O$_6$ is used in (**a, b, d–i**), and the datasets in (**d–i**) are averaged over 64 independent samples. To mimic the experimental procedures, the simulations (**d–i**) initiated with random states at $\mu_0H^z = -4.2$ T. We gradually raised $\mu_0H^z$ until reaching the target field and then simulated the relaxation processes in subsequent MCS. All the SSE-QMC calculations (**a, b, d–i**) were performed on a $6 \times 6 \times 12$ (432-site) cluster with periodic boundary conditions.

transverse fields ($0 < \Gamma < 0.02$ K, see Fig. 4). This discrepancy may arise from a low concentration of structural defects and internal transverse magnetic fields contributed by the dipole moments of Co, which are unavoidable in the real material and complexities not considered in our calculations. At 1.8 K, the quantum tunneling capacity of $\Gamma$ (indicated by the increase in $W_\downarrow/W_\uparrow$ due to $\Gamma$) is significantly reduced compared to that at 1 K (Fig. 4c). Consequently, the QA effects of $\Gamma$ at 1.8 K are expected to be notably weaker (Fig. 4g–i). Moreover, to mitigate the variability in the calculated MCS dependence of energy and magnetization, we averaged over 64 independent samples (despite the associated high computational cost). However, achieving complete elimination of this variability remains challenging in the MC simulations, especially at higher temperatures around 1.8 K, where thermal fluctuations come into play (Fig. 4g–i). Therefore, the main focus of this work is to present experimental findings regarding the QA effects of the small transverse field ($\Gamma \leq 0.1$ K) in $\alpha$-CoV$_2$O$_6$, and the numerical results only seek to provide a qualitative interpretation of the annealing effects of the transverse field at low temperatures (Fig. 4d–f).

Our large-scale classical MC simulations at $\Gamma = 0$ K demonstrate that $\alpha$-CoV$_2$O$_6$ exhibits long-lived metastable states at low temperatures, commonly in the form of KT phases containing topological

vortices, antivortices, and vortex-antivortex pairs (Supplementary Fig. 7)[6,18,23,24]. Moreover, we did not observe significant evidence supporting the presence of quenched structural defects that restrict the motion of domain walls. Instead, the vortices, antivortices, and vortex-antivortex pairs emerge around the domain walls, and are expected to decrease with increasing transverse field as indicated by changes in energies and longitudinal magnetization (Fig. 4d–f). Therefore, we suggest that these topological defects may play a crucial role in confining domain walls and increasing lifetimes of metastable states. However, it remains uncertain whether the applied transverse field directly affects the dynamics of domain walls or if it influences them indirectly through vortex dynamics, calling for further investigations.

## Heat transport measurements

The spin fluctuations around domain walls and topological defects can scatter carriers such as electrons[32] and phonons[33–36], and their properties may be studied through transport measurements. As $\alpha$-CoV$_2$O$_6$ is electrically insulating with a room temperature resistance larger than 20 MΩ, we performed heat transport measurements under transverse magnetic fields of $\mu_0H^y$ up to 8 T at $\mu_0H^z = 0$ T (Fig. 5, Supplementary Figs. 15, 16). At $\mu_0H^z = 0$ T, the QA effects of transverse

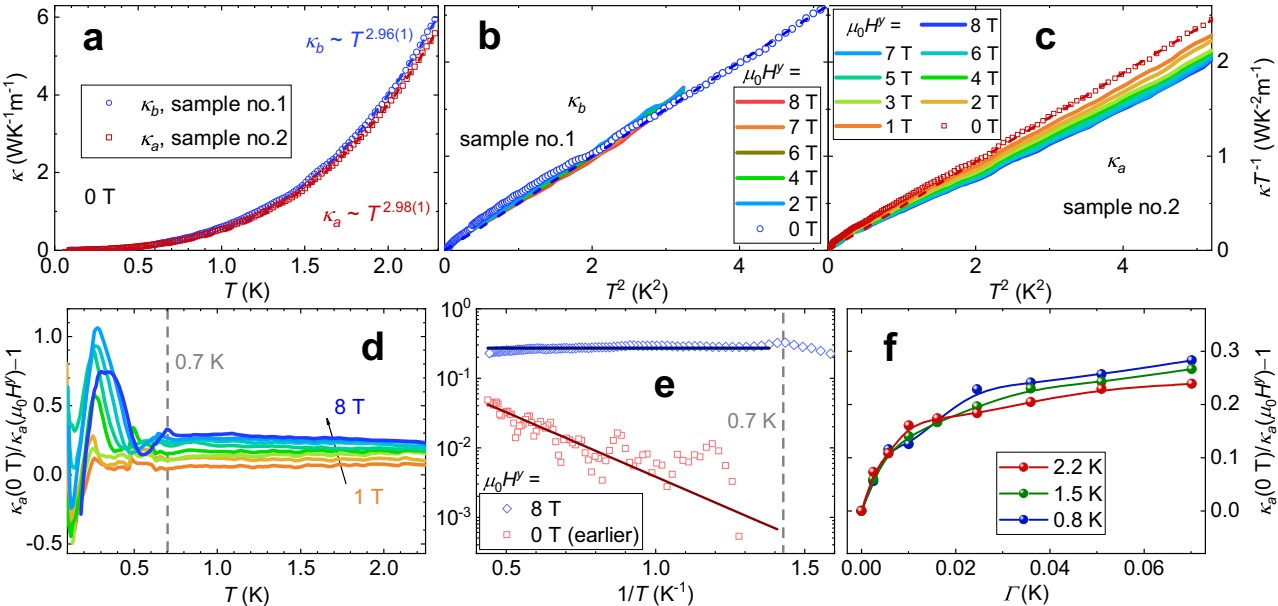

**Fig. 5 | Heat transport in transverse fields. a** Zero-field thermal conductivity of two α-CoV$_2$O$_6$ crystals measured with the heat flow along the *b*- and *a*-axis, $\kappa_b$ and $\kappa_a$, respectively. The dashed lines depict the power-law fits. **b**, **c** $\kappa_b$ and $\kappa_a$ measured under various transverse magnetic fields of $\mu_0 H^y$. The dashed lines in (**b**) and (**c**) show the $\kappa \propto T^3$ behaviors of the zero-field data. **d**, **e** Temperature dependence of $\kappa_a$(0 T)/$\kappa_a(\mu_0 H^y) - 1$. In (**e**), the zero-field data measured earlier is used (Supplementary Note 5), and the red and blue lines display the Arrhenius and constant fits above ~ 0.7 K. **f** Transverse field ($\Gamma$) dependence of $\kappa_a$(0 T)/$\kappa_a(\mu_0 H^y) - 1$.

field are also evident in the SSE-QMC simulations at low temperatures (Fig. 4d). Below the Néel temperature of $T_N$ ~ 14 K, the magnetic specific heat of α-CoV$_2$O$_6$ exhibits a gaped behavior with a large spin gap of ~ 35 K (Supplementary Fig. 9), indicating that the density of states of magnetic excitations is extremely low. Thereby, the main carriers in the low-*T* thermal conductivity (*κ*) measurements should be phonons.

In the temperature range from ~ 1 to 2.3 K, the zero-field thermal conductivity can be fitted by a power law, $\kappa = \gamma T^{\alpha'}$ (Fig. 5a). For the measurements with heat flow along the *b* and *a* axes, we got $\alpha'$ = 2.96(1) and 2.98(1), $\gamma$ = 0.514(5) and 0.477(3) WK$^{-4}$m$^{-1}$ for sample nos. 1 and 2, respectively, consistent with the dominance of phonons. In α-CoV$_2$O$_6$, we expect the ideal phonon thermal conductivity $\kappa_p \sim \frac{nC_p\lambda_{pp}\bar{v}_p}{3N_A} \sim \gamma T^3$ below 2.3 K, where $C_p \sim \frac{12\pi^4 R T^3}{5\Theta_D^3}$ is the phonon specific heat with the Debye temperature $\Theta_D$ ~ 160 K estimated from fitting the experimental specific heat of nonmagnetic α-ZnV$_2$O$_6$, $\lambda_{pp}$ represents the mean free path, $\bar{v}_p \sim \frac{k_B\Theta_D}{h}(6\pi^2 n)^{-1/3}$ (~ 3140 m/s) is the average phonon velocity, and $n$ ~ 5 × 10$^{27}$ m$^{-3}$ the density of unit cells. Using the measured values of $\gamma$, we obtained $\lambda_{pp} = \bar{v}_p\tau_{pp}$ ~ 0.12 mm ($\tau_{pp}^{-1}$ ~ 30 MHz represents the pure phonon scattering rate including crystal boundary contributions), comparable to the smallest dimension of the crystal (~ 0.15 mm).

The thermal conductivity measured along the *a* axis ($\kappa_a$) decreases significantly with increasing transverse magnetic field ($\mu_0 H^y$) as shown in Fig. 5c. In comparison, the thermal conductivity along the *b* axis ($\kappa_b$) exhibits only a weak decrease with increasing $\mu_0 H^y$ (Fig. 5b). These observations are in line with the MC simulations that the spin system has much more domain walls and topological defects along the frustrated triangular plane (i.e., the *ac* plane) compared to those along the spin chain (i.e., the *b* axis), at low temperatures (Supplementary Fig. 7). Hence, the suppression of $\kappa_a$ by $\mu_0 H^y$ is likely related to the movements of domain walls and topological defects. Due to the significant difference in scale between the microscale distances between domain walls (or topological defects) and the macroscale wavelengths of acoustic phonons, phonon scattering by relatively static domain walls and topological defects is negligible in the zero-field heat transport at low temperatures (Fig. 5a). The transverse magnetic field is believed to induce the deconfinement of domain walls and topological defects by flipping nearby Ising spins, which in turn can increase the phonon-spin

scattering rate ($\tau_{ps}^{-1}$) and suppress $\kappa_a$ (when $\tau_{ps}^{-1}$ gets comparable to $\tau_{pp}^{-1}$). To simplify the analysis, we used the latest measurement of zero-field thermal conductivity, $\kappa_a$(0 T), to approximate the ideal phonon thermal conductivity $\kappa_p$ along the *a* axis, even though there are internal transverse magnetic fields present (approximately ± 0.2 T, as mentioned above) and slight derivations from the $T^3$ law below ~ 1 K (Fig. 5c). We obtained $\kappa_a$(0 T)/$\kappa_a(\mu_0 H^y) - 1 \sim (\lambda_{pp} - \lambda_p)/\lambda_p = \tau_{pp}/\tau_{ps}$, where $\lambda_p = \bar{v}_p\tau_p$ is the mean free path at $\mu_0 H^y$ and $\tau_p^{-1} = \tau_{pp}^{-1} + \tau_{ps}^{-1}$ is the total scattering rate[33]. The ratio of $\kappa_a$(0 T)/$\kappa_a(\mu_0 H^y) - 1$ approximately reflects $\tau_{ps}^{-1}$ induced by $\mu_0 H^y$ in α-CoV$_2$O$_6$, given the negligible temperature and magnetic field dependencies of $\tau_{pp}$ (and $\bar{v}_p$) below 2.3 K.

If $\tau_{ps}^{-1}$ is thermally activated, it is expected to follow the Arrhenius form $\tau_{ps}^{-1} \propto \exp(-\Delta E/T)$. We observed this behavior and obtained a barrier energy of $\Delta E(\mu_0 H^z = 0$ T) ~ 4.3 ± 0.3 K by analyzing two thermal conductivity measurements taken during different run sequences (run #1 and #2) at 0.7 < *T* < 2.3 K and 0 T in α-CoV$_2$O$_6$ (see Fig. 5e). In contrast, in a transverse magnetic field of $\mu_0 H^y \geq 1$ T, $\tau_{ps}^{-1}$, as measured by $\kappa_a$(0 T)/$\kappa_a(\mu_0 H^y) - 1$, is nearly independent of temperature down to ~ 0.7 K (Fig. 5d, e), indicating quantum tunneling of magnetic domain walls. Above ~ 0.7 K, the QA effect of the transverse field almost saturates at $\Gamma$ ~ 0.07 K, as shown in Fig. 5f. On the one hand, the applied transverse field facilitates annealing, reduces domain walls, and thus increases the thermal conductivity by decreasing $\tau_{ps}^{-1}$ at low temperatures. On the other hand, as aforementioned, the transverse field causes quantum fluctuations between states with $S_i^z = \pm 1/2$ resulting in the motion of magnetic domain walls, increases $\tau_{ps}^{-1}$, and decreases the thermal conductivity. The transverse field's positive and negative effects likely compete, resulting in the observed oscillation of $\kappa_a$(0 T)/$\kappa_a(\mu_0 H^y) - 1$ with temperature, below ~ 0.7 K (Fig. 5d). Further investigation is required to fully understand the intriguing behaviors of $\kappa_a$ observed at low temperatures (Fig. 5c–f).

The thermal conductivity ($\kappa_a$) measured at the same transverse magnetic field and temperature exhibits a weak increase in run #2 compared to run #1, following a long-term (> 3 days) annealing in $\mu_0 H^y$. This observation potentially suggests a decrease in the concentrations of domain walls and topological defects along the triangular plane, after the prolonged annealing process. However, the suppression of $\kappa_a$

by $\mu_0 H^p$ remains largely repeatable in run #2, suggesting that the domain walls and topological defects do not completely disappear even after the long-term annealing in $\mu_0 H^p$ (up to 8 T) between temperatures of 0.05 and 2.3 K.

## Discussion

At $\mu_0 H^z = 2$ T, it is noteworthy that a small transverse field of $\Gamma \sim 3.5$ mK is unable to fully anneal the entire magnetic crystal of $\alpha$-CoV$_2$O$_6$ to the lowest-energy configuration at low temperatures, as indicated by the slight deviation of long-time $M^z$ from the lowest-energy value of $g^z/6$ (Fig. 2c). The QMC simulation shows that the metastable states can have extremely long lifetimes, lasting up to > 400,000 SSE MCS (×100, see Fig. 4). The presence of vortex-antivortex pairs may be a key factor contributing to these long lifetimes (Supplementary Fig. 7f). Under nonzero transverse field, the QA effects manifest primarily through a significant reduction in the annealing time constant (Fig. 3) and a profound increase in $M^z$ at long times (Fig. 2c), compared to those observed under zero transverse magnetic field (Fig. 2a), at the same low temperature. If one wishes for the system to converge completely to the optimum state, it seems to be necessary to apply a stronger $\Gamma$ and to wait longer. For instance, at 1.8 K, the long-time $M^z$ measured at $\mu_0 H^x \sim 3.4$ T is closer to $g^z/6$ than at $\mu_0 H^x \sim 2$ T (Supplementary Fig. 14). However, even with increased waiting times up to 5 hours, $M^z$ did not show further convergence towards $g^z/6$ at $\mu_0 H^x \sim 2$ T and lower temperatures. Furthermore, under transverse fields up to $\Gamma \sim 0.07$ K, $\kappa_a(\mu_0 H^p)$ is significantly lower than $\kappa_a(0\ \text{T})$ at least within 0.7–2.3 K (Fig. 5), indicating an abundance of domain walls along the frustrated triangular plane even after the long-term annealing (see above). In the QMC simulations, achieving complete convergence to the lowest-energy configuration of the 432-site cluster is also extremely difficult at low temperatures. In most independent samples, increasing $\Gamma$ up to 0.1 K (corresponding to e.g. $\mu_0 H^p \sim 10$ T) still did not result in complete convergence to the lowest-energy configuration after ~ 300,000 SSE MCS (×100, see Fig. 4).

There are at least three possible reasons for the incomplete QA observed in $\alpha$-CoV$_2$O$_6$ at low temperatures. First, metastable KT phases are predicted to be widespread in frustrated Ising models on triangular lattices[6,29,37,38]. The presence of vortices, antivortices, and vortex-antivortex pairs may confine domain walls, and prolong the annealing times in the extremely long-time range as the system approaches its lowest-energy configuration. Our many-body MC simulations, conducted at $T \geq 1$ K for up to large MCS, have effectively captured this incomplete annealing phenomenon (Fig. 4). It can be expected that annealing this frustrated spin system will become even more challenging at larger scales and low temperatures. Second, the transverse field $\Gamma$ can weaken the lowest-energy "up-up-down" order, leading to a decrease in longitudinal magnetization $M^z$. However, at $\Gamma \leq 0.1$ K this reduction of $M^z$ is estimated to be less than ~ $2 \times 10^{-4} \mu_B$/Co (Fig. 4b) $-|\Delta M^z_\Gamma|$ as estimated using $\mathcal{H}_s$ (see above), which is much smaller than the experimental value of $g^z/6 - M^z(t \gg 10\ \text{s}) \sim 0.4\ \mu_B$/Co (see Fig. 2c). In this study, the transverse field of $\Gamma \leq 0.1$ K is considered too weak to significantly alter the lowest-energy configurations, which cannot account for the incomplete QA observed. Finally, while we did not find strong evidence supporting the presence of structural imperfections in the single crystals of $\alpha$-CoV$_2$O$_6$, it is still possible that the real-world compound contains a low concentration of quenched structural defects. These defects could potentially impede magnetic domain wall movement at low temperatures, making complete annealing difficult, even in the presence of transverse fields.

At low temperatures ($\ll T_N$), (incomplete) QA towards the lowest-energy configuration is clearly observed in magnetic relaxation measurements under a tiny transverse field applied to $\alpha$-CoV$_2$O$_6$. Experimental evidence of quantum tunneling of magnetic domain walls and topological defects is observed in both the relaxation rates of magnetization and the phonon-spin scattering rates reflected in heat transport

measurements along the frustrated triangular plane. The distinctive observation of QA effects induced by the small transverse field may stimulate further experimental investigations and large-scale ab initio simulations in this frustrated magnet and other relevant materials.

## Methods

In the magnetization ($M^z$) measurements, we tilted the single crystal to apply a transverse magnetic field with respect to the external field (**H**), determining the longitudinal and transverse magnetic fields for each fixed angle $\theta$ between the Ising ($z$) direction and **H**. We used five single crystals with $\theta \sim 0°$, 15°, 30°, 45°, 60° to measure $M^z$ above 1.8 K in a magnetic properties measurement system, and measured on two smaller crystals with $\theta \sim 0°$ and 45° below 1.85 K using a Faraday force magnetometer in a $^3$He–$^4$He dilution refrigerator[39,40] (Supplementary Fig. 13). The low-$T$ thermal conductivity measurements were conducted using a standard four-wire method in the dilution refrigerator[33]. We conducted unbiased SSE-QMC calculations, utilizing both local diagonal and Wolff cluster updates[28,29], to simulate the model on finite lattice sizes. Each SSE MCS involves one local diagonal update (attempting to either remove or insert a diagonal operator and traversing through the operator strings) along with a set number of Wolff cluster updates (sufficient to averagely visit all operators in the string). Due to frustration combined with a low transverse field or high longitudinal field, the SSE Wolff cluster algorithm is less efficient at low temperatures. To make the relaxation processes clear, we initiated all the simulations with random states at $\mu_0 H^z = -4.2$ T. Classical MC simulations indicate that the spin system should quickly relax to the fully inversely polarized state after only 3-4 MCS at $\mu_0 H^z \sim -4$ T, please refer to Supplementary Fig. 7h. For more details, please refer to the Supplementary Information.

## Data availability

The data generated within the main text are provided in the Source Data. Additional raw data are available from the corresponding authors upon request.

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

## Acknowledgements

The authors gratefully acknowledge Jun Li and Xiaochen Hong for their technical assistance in the MC computations and heat transport measurements, respectively. This work was supported by the National Key Projects for Research and Development of China (Grant Nos. 2023YFA1406500 (Y.L.) and 2022YFA1403900 (H.L.)), the National Natural Science Foundation of China (Nos. 12274153 (Y.L.), 12074135 (J.W.), 21875249 (C.H.), 12322403 (H.L.), 12347107 (H.L.)), the Fundamental Research Funds for the Central Universities (No. HUST: 2020kfyXJJS054 (Y.L.)), the Strategic Priority Research Program of the Chinese Academy of Sciences (Nos. XDB0500202 (H.L.) and XDB33020300 (H.L.)), and the Youth Innovation Promotion Association CAS (No. 2021004 (H.L.)). Y.C.W. acknowledges the support from Zhejiang Provincial Natural Science Foundation of China (No. LZ23A040003), and the support from the High-performance Computing Centre of Hangzhou International Innovation Institute of Beihang University.

## Author contributions

Y.L. planed and supervised the experiments. Y.Z., Z.M., and Y.L. collected the magnetization, heat transport, specific heat, and x-ray diffraction data. J.W. and Z.H. provided the single crystals of $\alpha$-CoV$_2$O$_6$. Z.M. synthesized and characterized the $\alpha$-ZnV$_2$O$_6$ samples. Y.C.W. performed the stochastic series expansion calculations. Y.L. and H.L. conducted the classical Monte Carlo simulations. Y.L., H.L., Y.C.W., and Y.Z. analysed the data and wrote the manuscript with comments from all co-authors. The manuscript reflects the contributions of all authors.

## Competing interests

The authors declare no competing interests.
