## [Peer Review File · Nature Communications]

Reviewers' Comments:

Reviewer #1:

Remarks to the Author:

Paper: Quantum annealing of a well-defined frustrated magnet

Authors: Y. Zhao, Z. Ma, Z. He, H. Liao, J. Wang and Y. Li

The authors report on the observation of quantum annealing phenomena (from a symmetry-frozen state) for a tiny transverse field in a single-crystal study of α -CoV₂O₆. Quick convergence towards the lowest-energy state configuration (confirmed by neutron diffraction measurements and exact diagonalization in the model) with a nearly temperature-independent fast relaxation in a tiny effective transverse field (of about 3.5 mK; in 10 seconds, from about 15 hours without the transverse field at a temperature 1 K). The study is quite comprehensive and the observation of quantum annealing in this new sample is potentially interesting and important. The presentation is also nice.

I have some queries though. In the low temperature phase, without any effective) transverse field, was there any indication of non-ergodicity (or replica symmetry breaking)? Did the model calculations show any such indication? And with application of transverse field (due to tunneling), does one get back ergodicity or replica symmetry (cf. [17])? Does one get, in your sample, any effective Almeida-Thouless line (indicating replica symmetry breaking) of the longitudinal field dependent glass transition temperature in absence of the transverse field, and does that line disappear altogether when the tiny transverse field applied (similar to that shown by Schindler, Guaita, Shi, Demler and Cirac, PRL 2022, numerically: <https://journals.aps.org/prl/pdf/10.1103/PhysRevLett.129.220401>)

I can understand, these are not easy to show experimentally and in that case, maybe the authors can just address those points and put their comments. However if possible to show positively (or even address numerically in the supplementary note), the authors can perhaps remove the term 'apparent' from "We observe apparent quantum annealing phenomena" in the abstract of their paper.

Reviewer #2:

Remarks to the Author:

This study reports the relaxation dynamics of the multidomain magnetic state in the frustrated Ising ferromagnet α -CoV₂O₆. The authors found that the relaxation of multidomain magnetic state was significantly enhanced in the presence of the transverse magnetic field. The observed relaxation time under the transverse field was nearly temperature-independent, indicating the realization of quantum annealing. These experimental observations were supported by Monte Carlo simulations. Thermal transport properties were also measured under the transverse field and discussed in the light of the deconfined domain walls/topological defects. This is an interesting study, but I don't think the systematics of the experimental results has been fully investigated.

1. In Fig. 3a, the data points at 2 T appear to be quite exceptional, but there is no explanation as to why the quantum annealing is clearly seen only at 2 T.
2. In Fig. 3b, although the temperature-dependent and -independent profiles are contrasting and interesting, the so-called quantum-classical crossover is not demonstrated. I think the data between $\mu_0 H^x = 0$ T and 2 T need to be examined in more detail because such data are important in clarifying how the classical annealing changes to the quantum annealing. In fact, in the pioneering work (ref. 16), such quantum-classical crossover behaviour was presented and explicitly discussed.
3. This study discusses thermally activated or quantum annealing processes on the relaxation of the multi-domain state of the 1/3-plateau phase into the single-domain state. Thus, the mobility of the domain wall seems to be crucial in understanding the dynamics of this relaxation, while the authors also mention the dynamics of topological defects such as vortices and antivortices. Since

there is no microscopic evidence for the relationship between these two kinds of dynamics, it is not clear whether the transverse magnetic field has a direct effect or an indirect effect through vortex dynamics with respect to the domain-wall dynamics.

4. Although the authors argue that the deconfinement of the topological defects and domain walls would suppress the thermal conductivity, the relationship is still not convincing. I think that the comparison of the thermal conductivity before and after the annealing would be helpful to understand this issue. If possible, the time-dependent profile of the thermal conductivity during the thermal annealing would be more interesting. If the thermal conductivity is suppressed only when the topological defects and domain walls are deconfined, the suppression of the thermal conductivity would only be observed during the annealing process.

5. In Fig. 2f, the curve of $H^x = 1.7 H^z$ is missing.

6. In Fig. 2b, the value of the exponent of the stretched exponential behaviour should be explicitly mentioned. Then, whether the distribution of the relaxation time is small should be discussed.

In summary, although this study reports interesting results, there remain several important issues that need to be clarified. I think that these issues should be carefully addressed for this paper to be published in Nature Communications.

Reviewer #3:

Remarks to the Author:

The authors study spin-texture relaxation in α -CoV₂O₆ due to transverse magnetic fields by combining experiments and classical and quantum Monte Carlo simulations.

The analysis of the atomic limit is very interesting, where the fact that Co²⁺ has a Kramers doublet ground state is emphasized and thereby the authors indicate that strong easy-axis anisotropy (small transverse components of the g-tensor) does not preclude quantum effects. Furthermore, the relaxation of the magnetization process at 2T seems to demonstrate the remarkable acceleration of spin relaxation presumably due to the transverse magnetic field (Fig. 2). However, somehow, the authors refer to this as “apparent” quantum annealing phenomena, which is somewhat perplexing. In any case, my specialty is in theory and numerical calculations, so I will refrain from further commenting on the experiment.

In their model construction and quantum Monte Carlo simulation for analyzing many-body effects, I found many issues. First of all, the proposed reduction of the 3D system into the 2D model is very problematic. Of course, such a procedure is valid when the classical ground states are concerned and every state can be assumed to be ferromagnetically ordered in the chain direction. However, this is a different story when it comes to a quantum Hamiltonian for discussing quantum ground states and the time-dependent phenomena. Moreover, their procedure (from Eq. 1 to Eq. 2) for the quantum-classical mapping has an error, as discussed in the following.

As for the 3D-to-2D reduction, according to the main text, each spin in the effective 2D model (1) represents an arbitrary spin in a specified chain. However, this interpretation clearly does not correspond to the reality. For example, when a “representative” spin is flipped by a transverse field, this spin is most likely not directly coupled to another arbitrarily chosen spin in the neighboring chain. Therefore, the energy change in the interchain interaction is almost always uncaptured by the Ising term in the Hamiltonian. Another possible interpretation of Eq. (1), though not mentioned in the paper, could be a perturbative one, where J_0 is regarded as a non-perturbative term and each spin represents a doubly degenerate state (i.e., all-up or all-down) of a given chain. However, the transverse field in such an interpretation should be an effective one, since connecting all-up/down states requires higher-order perturbative spin flips of the entire chain. The strength of the effective transverse field cannot be the same as the original transverse field per spin.

In the quantum-classical mapping, the error lies in the fact that the rescaling is made neither in the temperature nor in the interchain interaction. To see this, we consider dividing the quantum Hamiltonian in d -dimensions as $H = H_0 + H_1$ with the Ising interaction term H_0 and the transverse

magnetic field term H_1 . Then, the first step of the mapping is to consider the Boltzmann density matrix at (actual) temperature T as

$$\exp(-H / T) = [\exp(-(H_0 + H_1) / (N_b T))]^{N_b} \sim [\exp(-H_0 / (N_b T)) \exp(-H_1 / (N_b T))]^{N_b},$$

which is then followed by insertion of complete basis sets in-between. Thus, in the effective $(d+1)$ -dimensional classical Hamiltonian, the temperature (or the interchain interaction) must be properly rescaled by the total Trotter number N_b (or $1 / N_b$).

Unfortunately, these serious issues make it almost impossible to consider proper comparison against the experiments, and for this reason, I do not recommend the publication of this work in nature communications.

Below, I list additional comments for the consideration by the authors:

1) In the simulation shown in Fig. 4, why not use the same initial condition and follow the same protocol as in the experiment?

2) On p. 11 of the SI, m_I is not an "order parameter" as it is always nonzero for nonzero transverse field. For the same reason, the system does not exhibit a "spontaneous" transverse magnetization along the x-axis. The transverse magnetization is simply induced by the external field.

Reviewer #1 (Remarks to the Author):

The authors report on the observation of quantum annealing phenomena (from a symmetry-frozen state) for a tiny transverse field in a single-crystal study of α -CoV₂O₆. Quick convergence towards the lowest-energy state configuration (confirmed by neutron diffraction measurements and exact diagonalization in the model) with a nearly temperature-independent fast relaxation in a tiny effective transverse field (of about 3.5 mK; in 10 seconds, from about 15 hours without the transverse field at a temperature 1 K). The study is quite comprehensive and the observation of quantum annealing in this new sample is potentially interesting and important. The presentation is also nice.

I have some queries though. In the low temperature phase, without any effective) transverse field, was there any indication of non-ergodicity (or replica symmetry breaking)? Did the model calculations show any such indication? And with application of transverse field (due to tunneling), does one get back ergodicity or replica symmetry (cf. [17])? Does one get, in your sample, any effective Almeida-Thouless line (indicating replica symmetry breaking) of the longitudinal field dependent glass transition temperature in absence of the transverse field, and does that line disappear altogether when the tiny transverse field applied (similar to that shown by Schindler, Guaita, Shi, Demler and Cirac, PRL 2022, numerically: <https://journals.aps.org/prl/pdf/10.1103/PhysRevLett.129.220401>)

I can understand, these are not easy to show experimentally and in that case, maybe the authors can just address those points and put their comments. However if possible to show positively (or even address numerically in the supplementary note), the authors can perhaps remove the term 'apparent' from "We observe apparent quantum annealing phenomena" in the abstract of their paper.

Reply to Comment 1.1: We extend our gratitude to the Reviewer for their meticulous evaluation and positive assessment of our work.

α -CoV₂O₆, unlike spin-glass systems, does not exhibit any noticeable distribution of interactions or interaction randomness. Instead, the observed out-of-equilibrium dynamics in this material is attributed to the geometrically frustrated (antiferromagnetic) interchain interactions within the triangular plane and the ferromagnetic intrachain interactions. In α -CoV₂O₆, no indication of replica symmetry breaking or glass transition was reported; instead, only a long-range Néel transition was observed at $T_N \sim 14$ K [e.g., J. Am. Chem. Soc. 131, 7554 (2009); Phys. Rev. B 86, 214428 (2012); Phys. Rev. B 87, 024403 (2013)].

α -CoV₂O₆ undergoes a Néel transition at $T_N \sim 14$ K, as confirmed by quasi-equilibrium-state specific heat (please see Fig. R3.2b), magnetization (Fig. R3.2d), and the previously reported neutron diffraction measurements [Phys. Rev. B 86, 214428 (2012)]. Our Monte Carlo calculations effectively simulate this Néel transition (please see Fig. R3.2b,d and Supplementary Fig. 6b). The transition temperature T_N is slightly suppressed by the application of a transverse magnetic field up to $\mu_0 H^y = 9$ T (please see

Supplementary Fig. 2a,b), corresponding to a small effective transverse field of $\Gamma \sim 0.1$ K. However, achieving complete suppression of the ground-state Néel ordering necessitates a higher effective transverse field of $\Gamma \sim 9$ K, as estimated by the exact diagonalization calculations (Supplementary Fig. 10b). Reaching $\Gamma \sim 9$ K would require an excessively high transverse magnetic field of $\mu_0 H^y \sim 40$ T, leading to significant stress within the single crystal and potential sample destruction due to the strong Ising magnetic anisotropy [Phys. Rev. B 97, 184434 (2018)]. As anticipated by the Reviewer, experimentally restoring symmetry by a transverse field is challenging due to the relatively strong spin-spin couplings in α -CoV₂O₆. The primary focus of this work is to investigate the quantum annealing effect resulting from the presence of a tiny transverse field.

Fig. R1.1 | Longitudinal magnetic field dependence of quasi-equilibrium-state magnetization, both in the absence and presence of small transverse fields. The experimental data (expt.) is represented by colored scatter points, while the colored lines depict the corresponding Monte Carlo calculations (calc.).

Under quasi-equilibrium-state conditions, the frustrated magnet α -CoV₂O₆ exhibits two longitudinal-field-induced magnetic transitions at low temperatures. These transitions occur as the longitudinal magnetic field increases at low temperatures, transforming the system from a stripe antiferromagnetic state to a 1/3 magnetization plateau state and eventually to a fully polarized phase (see Fig. R1.1). The application of a small transverse field does not eliminate the longitudinal-field-induced transitions, which is in good agreement with calculations based on the microscopic model (see Fig. R1.1). Our experimental study on the frustrated magnet α -CoV₂O₆, which lacks evident interaction randomness, may bring forth new interest, as most previous works have focused on discussing quantum annealing effects in quantum Ising spin-glass systems. The revised manuscript cites the relevant and intriguing work on the transverse- and longitudinal-field evolutions of the effective Almeida-Thouless line [Phys. Rev. Lett. 129, 220401 (2022)].

Furthermore, we have incorporated the Reviewer's suggestion and removed the word "apparent" from the revised manuscript.

Reviewer #2 (Remarks to the Author):

This study reports the relaxation dynamics of the multidomain magnetic state in the frustrated Ising ferromagnet α -CoV₂O₆. The authors found that the relaxation of multidomain magnetic state was significantly enhanced in the presence of the transverse magnetic field. The observed relaxation time under the transverse field was nearly temperature-independent, indicating the realization of quantum annealing. These experimental observations were supported by Monte Carlo simulations. Thermal transport properties were also measured under the transverse field and discussed in the light of the deconfined domain walls/topological defects. This is an interesting study, but I don't think the systematics of the experimental results has been fully investigated.

1. In Fig. 3a, the data points at 2 T appear to be quite exceptional, but there is no explanation as to why the quantum annealing is clearly seen only at 2 T.

Reply to Comment 2.1: We appreciate the Reviewer's interest in our work and his/her valuable comments.

Classical Monte Carlo (MC) simulations of α -CoV₂O₆ reveal that the densities of domain walls and topological defects (including vortices, antivortices, and vortex-antivortex pairs) are maximized at a longitudinal field of $\mu_0 H^z \sim 2$ T, after the same number of MC steps (annealing time, please see Supplementary Fig. 7). The calculated relaxation time, measured in MC steps, also exhibits a peak at $\mu_0 H^z \sim 2$ T, consistent with the experimental findings (please see Supplementary Fig. 8c). At this field strength, slightly above the critical field $\mu_0 H_{c1}^z \sim 1.5$ T, the Zeeman interaction drives the frustrated spin system close to a critical point, transitioning from the stripe antiferromagnetic state to the 1/3 magnetization plateau phase. In the absence of a transverse field, this configuration results in the highest concentrations of domain walls and topological defects, along with the longest relaxation time. Consequently, quantum annealing is most clearly seen at $\mu_0 H^z = 2$ T when a transverse field is applied.

We have incorporated the above discussion into the revised manuscript as well.

2. In Fig. 3b, although the temperature-dependent and -independent profiles are contrasting and interesting, the so-called quantum-classical crossover is not demonstrated. I think the data between $\mu_0 H^x = 0$ T and 2 T need to be examined in more detail because such data are important in clarifying how the classical annealing changes to the quantum annealing. In fact, in the pioneering work (ref. 16), such quantum-classical crossover behaviour was presented and explicitly discussed.

Reply to Comment 2.2: We greatly appreciate the Reviewer's professional suggestion. In response, we conducted measurements on two additional samples with $\theta \sim 15^\circ$ and 30° using a magnetic properties measurement system (MPMS), where θ is the angle between the Ising (z) direction and the applied magnetic field \mathbf{H} . We have consolidated all the MPMS data in Fig. R2.1, and the quantum-classical crossover

behavior is now clearly visible.

Fig. R2.1 | Magnetic properties of α -CoV₂O₆ measured with $\theta \sim 0^\circ, 15^\circ, 30^\circ, 45^\circ,$ and 60° in a MPMS. **a** M^z - H^z hysteresis loops measured at 1.8 K under different applied transverse fields (H^x) with respect to the longitudinal field (H^z), $H^x = \tan \theta H^z$. **b** The loop widths of the curves in **a**. **c** The relaxation time τ measured under $H^x = \tan \theta H^z$ at 1.8 K. **d** Inverse-temperature dependence of the relaxation rate τ^{-1} at the longitudinal field of $\mu_0 H^z = 2$ T and various transverse magnetic field of $\mu_0 H^x \sim 0, 0.5, 1.2, 2, 3.4$ T. θ represents the angle between the Ising (z) direction and the applied magnetic field \mathbf{H} , and the colored arrows indicate the peak temperatures. The low- T data measured at $\mu_0 H^x \sim 0$ T (hollow squares) and 2 T (hollow triangles) in a dilution refrigerator are shown for comparison. The thick blue line displays the Arrhenius behavior of $\tau^{-1} \propto \exp(-\Delta E/T)$.

The loop width is progressively suppressed as the transverse magnetic field increases, at $\mu_0 |H^z| > \mu_0 H_{c1}^z \sim 1.5$ T (see Fig. R2.1a,b). Additionally, the transverse magnetic field gradually decreases the relaxation time at $\mu_0 H^z = 2$ T and low temperatures (see Fig. R2.1c,d). The transverse magnetic field gradually lowers the peak temperature below which the Arrhenius behavior of $\tau^{-1} \propto \exp(-\Delta E/T)$ becomes visible (see Fig. R2.1d). The revised manuscript includes the newly acquired data and explicitly discusses the classical-quantum crossover behavior induced by the transverse magnetic field.

3. This study discusses thermally activated or quantum annealing processes on the relaxation of the multi-domain state of the 1/3-plateau phase into the single-domain state. Thus, the mobility of the domain

wall seems to be crucial in understanding the dynamics of this relaxation, while the authors also mention the dynamics of topological defects such as vortices and antivortices. Since there is no microscopic evidence for the relationship between these two kinds of dynamics, it is not clear whether the transverse magnetic field has a direct effect or an indirect effect through vortex dynamics with respect to the domain-wall dynamics.

Reply to Comment 2.3: We concur with the Reviewer’s viewpoint. Our current Monte Carlo simulations, utilizing the effective model, demonstrate that the application of a tiny transverse field considerably reduces the concentrations of both domain walls and topological defects after the same number of MC steps (annealing time, see Fig. 4 in main text and Supplementary Fig. 12). However, it remains uncertain whether the applied transverse field directly affects the dynamics of domain walls or if it influences them indirectly through vortex dynamics, calling for further investigations. Our experimental findings, coupled with the spin Hamiltonian with well refined parameters, serve as a promising starting point for future in-depth studies.

The incomplete explanation has been addressed and rectified in the revised manuscript.

4. Although the authors argue that the deconfinement of the topological defects and domain walls would suppress the thermal conductivity, the relationship is still not convincing. I think that the comparison of the thermal conductivity before and after the annealing would be helpful to understand this issue. If possible, the time-dependent profile of the thermal conductivity during the thermal annealing would be more interesting. If the thermal conductivity is suppressed only when the topological defects and domain walls are deconfined, the suppression of the thermal conductivity would only be observed during the annealing process.

Reply to Comment 2.4: We appreciate the reviewer’s interesting suggestion. However, measuring the short-time dependence of thermal conductivity (κ) poses challenges. The need to wait approximately 10 minutes for the thermometers to reach thermal equilibrium in each measurement hinders accurate real-time measurements. Additionally, the resistances of RuO₂ chip thermometers (RX-102A-BR) exhibit weak time and magnetic-field dependence, requiring *in-situ* calibration by turning off the heater for each measurement. To our knowledge, there are few reports on the measurements of (short-)time-dependent thermal conductivities at a temperature below ~ 2 K and under a magnetic field using the standard four-wire steady-state method.

Nevertheless, our low-temperature data still provide a rough reflection of the temporal variation of thermal conductivity (κ_a) over a longer time scale of approximately 3-10 days (please see Fig. R2.2a).

Let us briefly reiterate our understanding of the observed thermal conductivity. In comparison to the thermal conductivity along the spin chain (κ_b , see Fig. R2.2b), the thermal conductivity along the triangular plane (κ_a , see Fig. R2.2a) exhibits a distinct suppression under a transverse magnetic field. At low temperatures ($T \leq 4$ K), our Monte Carlo simulations and the previously reported neutron diffraction results [Phys.

Fig. R2.2 | Comparison of thermal conductivity (κ) measurements in $\alpha\text{-CoV}_2\text{O}_6$ along the a (κ_a , **a**) and b (κ_b , **b**) axes under various transverse magnetic fields ($\mu_0 H^y$). In run #1, κ_a was measured between ~ 0.06 and 2.3 K (measured by warming up) under steady $\mu_0 H^y$ ranging from 0 to 8 T. Each $\mu_0 H^y$ measurement took 1 day, resulting in a total duration of approximately 10 days for run #1, including cooling-down and liquid-helium-filling intervals. Following the completion of run #1, we cooled down to the base temperature of ~ 0.05 K, set $\mu_0 H^y$ to 0 T, and subsequently initiated the measurements for run #2 in sequence.

Rev. B 86, 214428 (2012)] indicate that the Ising spins are predominantly ferromagnetically ordered along each chain, protected by a sizable energy gap of $|J_0| \sim 30$ K (confirmed by specific heat measurements). Consequently, magnetic imperfections, including domain walls and defects, primarily exist along the frustrated triangular plane. The observed suppression of κ_a by the transverse magnetic field $\mu_0 H^y$ is most likely associated with these magnetic imperfections. The transverse field is expected to induce spin flips around these imperfections and increase spin fluctuations along the frustrated triangular plane, potentially accounting for the suppression of κ_a by $\mu_0 H^y$. The observation of temperature-independent (or thermal-independent) spin fluctuations along the frustrated triangular plane, as measured by $\kappa_a(0\text{ T})/\kappa_a(\mu_0 H^y) - 1$ at low temperatures (see main text for details), supports the quantum origin and strengthens the above explanations.

As anticipated by the Reviewer, the thermal conductivity (κ_a) measured at the same transverse

magnetic field and temperature indeed exhibits a weak increase in run #2 compared to run #1, following a long-term exploration in $\mu_0 H^y$ (Fig. R2.2a). This phenomenon is consistently observed across different transverse magnetic fields of $\mu_0 H^y = 0, 6, 7, 8$ T (refer to Fig. R2.2a). These observations potentially suggest a decrease in the concentrations of magnetic domain walls and defects along the triangular plane in run #2 compared to run #1, following the prolonged annealing period.

However, the suppression of κ_a by $\mu_0 H^y$ remains largely repeatable in run #2 (Fig. R2.2a), suggesting that the magnetic imperfections do not completely disappear even after the long-term (> 1 day) exploration in $\mu_0 H^y$ (up to 8 T) between temperatures of 0.05 and 2.3 K. **We kindly remind the Reviewer that the annealing process of the frustrated spin system α -CoV₂O₆ towards the ground state continues even after long-term annealing at a transverse field of $T < 0.1$ K. This implies that the ideal single-domain state is never achieved experimentally in α -CoV₂O₆ at low temperatures and in $T < 0.1$ K.** The measured magnetization (M^z) does not reach the expected value of $g^z/6 = 1.8 \mu_B/\text{Co}$, even after a prolonged annealing in the transverse field at low temperatures ($M^z \leq 1.4 \mu_B/\text{Co}$, see Fig. 2 in the main text). Furthermore, our quantum MC simulations indicate that metastable states of the 28,800-site system can have extremely long lifetimes at $T < 0.1$ K, lasting up to $\sim 300,000$ MC steps ~ 30 days (see Fig. 4 in main text and Supplementary Fig. 12d). Therefore, complete annealing of the quasi-infinite system of α -CoV₂O₆ in the thermodynamic limit (at $T < 0.1$ K) would require even more time.

The relaxation processes in the time scale of ~ 0.1 -50,000 s have been thoroughly investigated through magnetization measurements (refer to Fig. 2 in main text and Supplementary Fig. 14). These measurements enable the capture of the real-time pure intrinsic signal of the spin system. We did not attempt the uncommon measurements of the time dependence of thermal conductivity in our wet dilution refrigerator for the resubmission. As we have not previously conducted such measurements, we cannot guarantee their success. Furthermore, we believe that most of the Reviewer's concerns have been addressed by repeating the measurements (Fig. R2.2, as discussed above). We express our gratitude to the Reviewer once again for his/her helpful comments. The above discussion has been incorporated into the revised manuscript.

5. In Fig. 2f, the curve of $H^x = 1.7 H^z$ is missing.

Reply to Comment 2.5: The data for $H^x \sim 1.7 H^z$ ($\theta \sim 60^\circ$) at temperatures ranging from 0.08 K to 1.8 K were not measured in the dilution refrigerator. Our milli-Kelvin magnetization measurements were conducted solely on two single-crystal samples with angles $\theta \sim 0^\circ$ and 45° , where θ represents the angle between the Ising (z) direction and the applied magnetic field (\mathbf{H}). Consequently, at 0.5 K, only the data for $H^x (= \tan \theta H^z) \sim 0$ ($\theta \sim 0^\circ$, for thermal annealing) and $H^x \sim H^z$ ($\theta \sim 45^\circ$, for quantum annealing) are available.

Recently, we have followed the Reviewer's suggestion and conducted magnetization measurements on

two additional samples with angles $\theta \sim 15^\circ$ and 30° in MPMS, down to 1.8 K, to observe the classical-quantum crossover (please refer to Reply to Comment 2.2). To ensure clarity, in Fig. 2 of the revised main text, we only present the data obtained from samples with angles $\theta \sim 0^\circ$ and 45° . The magnetization data measured in MPMS on samples with angles $\theta \sim 0^\circ, 15^\circ, 30^\circ, 45^\circ,$ and 60° are displayed together for comparison in Fig. R2.1, included in the Supplementary Information.

6. In Fig. 2b, the value of the exponent of the stretched exponential behaviour should be explicitly mentioned. Then, whether the distribution of the relaxation time is small should be discussed.

In summary, although this study reports interesting results, there remain several important issues that need to be clarified. I think that these issues should be carefully addressed for this paper to be published in Nature Communications.

Reply to Comment 2.6: The measured stretching exponents of β are displayed in the inset of Supplementary Fig. 14d, ranging from ~ 0.5 to ~ 0.9 at $T \leq 2$ K. These experimental values are roughly consistent with the Monte Carlo (MC) calculations (refer to Supplementary Fig. 8b). In the MC simulations, the relaxation processes of independent samples have varying durations (see Supplementary Fig. 8a), resulting in a substantial distribution of relaxation times and thus displaying a pronounced stretched-exponential relaxation behavior. This discussion has been included in the revised manuscript.

We would like to express our gratitude to the Reviewer once again for the recommendation and all the helpful comments. We sincerely hope that the Reviewer is satisfied with our response, as well as the revisions made in the manuscript.

Reviewer #3 (Remarks to the Author):

The authors study spin-texture relaxation in α -CoV₂O₆ due to transverse magnetic fields by combining experiments and classical and quantum Monte Carlo simulations. The analysis of the atomic limit is very interesting, where the fact that Co²⁺ has a Kramers doublet ground state is emphasized and thereby the authors indicate that strong easy-axis anisotropy (small transverse components of the g-tensor) does not preclude quantum effects. Furthermore, the relaxation of the magnetization process at 2T seems to demonstrate the remarkable acceleration of spin relaxation presumably due to the transverse magnetic field (Fig. 2). However, somehow, the authors refer to this as apparent quantum annealing phenomena, which is somewhat perplexing. In any case, my specialty is in theory and numerical calculations, so I will refrain from further commenting on the experiment.

Reply to Comment 3.1: We extend our gratitude to the Reviewer for his/her meticulous reading of our manuscript and providing us with highly professional comments. As our expertise lies primarily in experimental work, we sincerely apologize for the mistakes made previously in the representation/derivation of the effective model. In this point-to-point response and the revised manuscript, we have taken great care to address the issues raised by the Reviewer. We hope that the Reviewer is satisfied with our response, as well as the revisions made in the manuscript. Furthermore, we have removed the word “apparent”.

In their model construction and quantum Monte Carlo simulation for analyzing many-body effects, I found many issues. First of all, the proposed reduction of the 3D system into the 2D model is very problematic. Of course, such a procedure is valid when the classical ground states are concerned and every state can be assumed to be ferromagnetically ordered in the chain direction. However, this is a different story when it comes to a quantum Hamiltonian for discussing quantum ground states and the time-dependent phenomena. Moreover, their procedure (from Eq. 1 to Eq. 2) for the quantum-classical mapping has an error, as discussed in the following. As for the 3D-to-2D reduction, according to the main text, each spin in the effective 2D model (1) represents an arbitrary spin in a specified chain. However, this interpretation clearly does not correspond to the reality. For example, when a representative spin is flipped by a transverse field, this spin is most likely not directly coupled to another arbitrarily chosen spin in the neighboring chain. Therefore, the energy change in the interchain interaction is almost always uncaptured by the Ising term in the Hamiltonian. Another possible interpretation of Eq. (1), though not mentioned in the paper, could be a perturbative one, where J_0 is regarded as a non-perturbative term and each spin represents a doubly degenerate state (i.e., all-up or all-down) of a given chain. However, the transverse field in such an interpretation should be an effective one, since connecting all-up/down states requires higher-order perturbative spin flips of the entire chain. The strength of the effective transverse field cannot be the same as the original transverse field per spin.

In the quantum-classical mapping, the error lies in the fact that the rescaling is made neither in the temperature nor in the interchain interaction. To see this, we consider dividing the quantum Hamiltonian in d -dimensions as $H = H_0 + H_1$ with the Ising interaction term H_0 and the transverse magnetic field term H_1 . Then, the first step of the mapping is to consider the Boltzmann density matrix at (actual) temperature T as

$$\exp(-H / T) = [\exp(-(H_0 + H_1) / (N_b T))]^{N_b} \sim [\exp(-H_0 / (N_b T)) \exp(-H_1 / (N_b T))]^{N_b},$$

which is then followed by insertion of complete basis sets in-between. Thus, in the effective $(d+1)$ -dimensional classical Hamiltonian, the temperature (or the interchain interaction) must be properly rescaled by the total Trotter number N_b (or $1 / N_b$).

Unfortunately, these serious issues make it almost impossible to consider proper comparison against the experiments, and for this reason, I do not recommend the publication of this work in nature communications.

Reply to Comment 3.2: First and foremost, we offer our sincere apologies for the errors made during the initial derivation of the effective model. However, we would like to highlight that the effective model remains valid and captures the experimental observations below ~ 4 K (please refer to the details provided below), in our setup.

Yes, we acknowledge that we used a perturbative method without considering higher-order effects of the transverse field at extremely low temperatures ($T \ll |J_0|$), based on the experimental fact that $\Gamma < 0.1$ K $\sim 0.003|J_0|$. In our experimental investigation of α -CoV₂O₆, the strength of the ferromagnetic intrachain exchange ($J_0 \sim -30$ K) is significantly larger than the maximum transverse field per spin ($\Gamma < 0.1$ K) and much larger than the antiferromagnetic interchain exchanges (~ 10 K) [also see Phys. Rev. B 87, 024403 (2013)]. No profound difference is observed between the quasi-equilibrium thermodynamic properties measured at $\Gamma = 0$ and $0 < \Gamma < 0.1$ K (refer to Fig. R3.1). The doubly degenerate ground states (all-up or all-down) of the pure ferromagnetic spin chain are well separated from other excited states by a large energy gap of $|J_0| \sim 30$ K. This gap value is confirmed by the specific heat measurements on α -CoV₂O₆, which remain almost independent of the small Γ (see Fig. R3.1a). At $T \leq 4$ K (i.e., $1/T \geq 0.25$ K⁻¹), the specific heat of α -CoV₂O₆ is extremely small and falls below the limit of measurability, indicating that most of the spin degrees of freedom are frozen along the ferromagnetic chains. This is consistent with our classical Monte Carlo (MC) simulations and the previously reported neutron diffraction results [Phys. Rev. B 86, 214428 (2012)], which demonstrate the nearly identical spin states of Co²⁺ ions along each chain. Below ~ 4 K, the spin system of α -CoV₂O₆ resides in a subspace of states with nearly identical spins along each chain, validating the effectiveness of the low- Γ effective model (further details provided below).

Fig. R3.1 | Quasi-equilibrium-state thermodynamic properties of α - CoV_2O_6 measured with and without applied transverse fields. **a** Temperature dependence of specific heat measured under various small transverse fields. Used $C_m \sim \exp(-\Delta/T)$, the gaps were fit to $\Delta = 35.3 \pm 0.3$ K and 35.0 ± 0.6 K at $\mu_0 H^x = 0$ T and 9 T, respectively. **b** Magnetization as a function of longitudinal field.

The original 3D Hamiltonian is formulated with the transverse field per spin,

$$\mathcal{H} = J_0 \sum_{\langle i, i_0 \rangle} S_i^z S_{i_0}^z + J_1 \sum_{\langle i, i_1 \rangle} S_i^z S_{i_1}^z + J_2 \sum_{\langle i, i_2 \rangle} S_i^z S_{i_2}^z + J_3 \sum_{\langle i, i_3 \rangle} S_i^z S_{i_3}^z - \mu_0 \mu_B H^z g^z \sum_i S_i^z - \Gamma \sum_i S_i^x. \quad (1)$$

In the subspace of states with nearly identical spins along each chain (all-up or all-down, and thus the spin operators S_i become independent of the site index along the chain), this 3D Hamiltonian is approximated as

$$\mathcal{H} \approx N_b (J'_1 \sum_{\langle I, I_1 \rangle} S_I^z S_{I_1}^z + J'_2 \sum_{\langle I, I_2 \rangle} S_I^z S_{I_2}^z + J'_3 \sum_{\langle I, I_3 \rangle} S_I^z S_{I_3}^z - \mu_0 \mu_B H^z g^z \sum_I S_I^z - \Gamma \sum_I S_I^x) + J_0 N/4, \quad (2)$$

with the length of spin chains N_b , the total number of spins N , and $J'_1 = 2J_1$, $J'_2 = J_2$, $J'_3 = 2J_3$. **Here, S_I is the spin operator acting on a “representative” Co^{2+} ion in the I th chain (i.e., acting on the spin state of the majority of Co^{2+} ions in the chain). As anticipated by the Reviewer, the strengths of both the transverse field and interchain couplings are effectively multiplied by N_b in Eq. (2).** The Hamiltonian described by Eq. (2) yields the same ground state as the reduced 2D Hamiltonian,

$$\mathcal{H}_{2D} = J'_1 \sum_{\langle I, I_1 \rangle} S_I^z S_{I_1}^z + J'_2 \sum_{\langle I, I_2 \rangle} S_I^z S_{I_2}^z + J'_3 \sum_{\langle I, I_3 \rangle} S_I^z S_{I_3}^z - \mu_0 \mu_B H^z g^z \sum_I S_I^z - \Gamma \sum_I S_I^x. \quad (3)$$

At $T \leq 4$ K, observables can be approximated as $\langle O \rangle = \frac{\text{Tr}(O e^{-\frac{\mathcal{H}}{k_B T}})}{\text{Tr}(e^{-\frac{\mathcal{H}}{k_B T}})} \approx \frac{\text{Tr}(O e^{-\frac{N_b \mathcal{H}_{2D}}{k_B T}})}{\text{Tr}(e^{-\frac{N_b \mathcal{H}_{2D}}{k_B T}})} = \frac{\text{Tr}(O e^{-\frac{\mathcal{H}_{2D}}{k_B T'}})}{\text{Tr}(e^{-\frac{\mathcal{H}_{2D}}{k_B T'}})}$, after eliminating the constant term $J_0 N/4$. **Here, $T' = T/N_b$ is the rescaled temperature due to the reduction from 3D to 2D, and it will be further rescaled to $T'' = T' N_{b'} = T N_{b'}/N_b$ when we return to another**

effective 3D model (see below), where $N_{b'}$ represents the number of Trotter replicas. Because we chose $N_{b'} = N_b = 50$, we got $T'' = T$ and did not find any problem in simulating the experimental data, in the previous manuscript.

Following the Reviewer's suggestion, we proceed with the derivation of the Boltzmann density matrix using Eq. (2) in the limit $T \rightarrow 0$ K (at $T \leq 4$ K),

$$\begin{aligned} \exp(-\mathcal{H}/(k_B T)) &\approx \exp(-J_0 N/(4k_B T)) [\exp(-\mathcal{H}_0/(k_B T' N_{b'})) \exp(-\mathcal{H}_1/(k_B T' N_{b'}))]^{N_{b'}} \\ &= C' \exp(-\mathcal{H}_{\text{eff}}/(k_B T'')), \end{aligned} \quad (4)$$

where $T'' = T' N_{b'} = T N_{b'}/N_b$ represents the final rescaled temperature, $\mathcal{H}_0 = J'_1 \sum_{\langle I, I_1 \rangle} S_I^z S_{I_1}^z + J'_2 \sum_{\langle I, I_2 \rangle} S_I^z S_{I_2}^z + J'_3 \sum_{\langle I, I_3 \rangle} S_I^z S_{I_3}^z - \mu_0 \mu_B H^z g^z \sum_I S_I^z$, $\mathcal{H}_1 = -\Gamma \sum_I S_I^x$, and

$$\mathcal{H}_{\text{eff}} = J'_1 \sum_{\langle I', I'_1 \rangle} S_{I'}^z S_{I'_1}^z + J'_2 \sum_{\langle I', I'_2 \rangle} S_{I'}^z S_{I'_2}^z + J'_3 \sum_{\langle I', I'_3 \rangle} S_{I'}^z S_{I'_3}^z - \mu_0 \mu_B H^z g^z \sum_{I'} S_{I'}^z + J'_0 \sum_{\langle I', I'_0 \rangle} S_{I'}^z S_{I'_0}^z. \quad (5)$$

In the above derivation, we have utilized

$$\exp\left(\frac{\Gamma S_I^x}{k_B T''}\right) = \cosh\left(\frac{\Gamma}{2k_B T''}\right) [\mathbf{I}_{2 \times 2} + 2 \tanh\left(\frac{\Gamma}{2k_B T''}\right) S_I^x] \quad (6)$$

and

$$\exp\left(\frac{-J'_0 S_{I'}^z S_{I'_0}^z}{k_B T''}\right) = \exp\left(\frac{-J'_0}{4k_B T''}\right) \begin{pmatrix} 1 & e^{\frac{J'_0}{2k_B T''}} \\ e^{\frac{J'_0}{2k_B T''}} & 1 \end{pmatrix} = \exp\left(\frac{-J'_0}{4k_B T''}\right) [\mathbf{I}_{2 \times 2} + 2 \exp\left(\frac{J'_0}{2k_B T''}\right) S_I^x], \quad (7)$$

where $\mathbf{I}_{2 \times 2}$ is the 2×2 identity matrix. Therefore, the ‘‘ferromagnetic coupling’’ in the Trotter direction $J'_0 = 2k_B T'' \ln[\tanh(\frac{\Gamma}{2k_B T''})]$ is obtained as depicted in Fig. R3.2a, and the low- T (≤ 4 K) magnetic properties of α -CoV₂O₆ can be effectively described by the Hamiltonian in Eq. (5). Moreover, the constant pre-factor $C' = \exp(\frac{N J'_0 - N J_0}{4k_B T}) [\cosh(\frac{\Gamma}{2k_B T''})]^{N_{ac} N_{b'}}$ is automatically eliminated when calculating observables, where N_{ac} represents the number of spin chains. **Because we set $N_{b'} = N_b = 50$, the final rescaled temperature is equal to the actual temperature, $T'' = T$.**

In the effective 3D model, such as with $\Gamma = 0.1$ K, the strength of (ferromagnetic) J'_0 increases with rising temperature (becomes unreasonably large at high temperatures, see Fig. R3.2a), the intrachain spin degrees of freedom remain frozen, and thus the calculated specific heat remains negligible up to ~ 30 K (see Fig. R3.2b). Consequently, the effective 3D model of Eq. (5) fails to accurately describe the magnetism of α -CoV₂O₆ above ~ 4 K (see Fig. R3.2b-d). To simulate the magnetic properties above ~ 4 K, one can revert to the original 3D model of Eq. (1) with zero transverse field ($\Gamma = 0$ K), as the quantum effects of the small Γ on both quasi-equilibrium (see Fig. R3.1) and dynamic (see Fig. R2.1d) properties are

Fig. R3.2 | Simulations of the quasi-equilibrium-state thermodynamic properties of α - CoV_2O_6 . **a** Comparison of the original ferromagnetic intrachain coupling J_0 with $J'_0(\Gamma) = 2k_B T'' \ln[\tanh(\frac{\Gamma}{2k_B T''})]$. **b, c, d** Calculations of magnetic specific heat, magnetization, and susceptibility using the original 3D Hamiltonian [Eq. (1)] with $\Gamma = 0$ K, and the effective Hamiltonian of Eq. (5) with J'_0 ($\Gamma = 0.1$ K) and with J'_0 fixed to J_0 . Experimental data are represented by scatter plots for comparison, and the same set of refined interaction parameters (as described in the manuscript) is used.

negligible. Alternatively, one can use the effective 3D model of Eq. (5) by setting $J'_0 \equiv J_0$ (Fig. R3.2b-d), which effectively captures the release of intrachain spin degrees of freedom in this high-temperature range. As shown in Fig. R3.2b-d, the consistency between the black (original) and blue (effective) curves indicates that the ferromagnetic correlations along the spin chain remain strong, and the differences in detailed interchain interaction configurations between the original and effective 3D models have minimal effect at $T < |J_0| \sim 30$ K.

The energy gap extracted from the experimental specific heat (Fig. R3.1a), $\Delta \sim |J_0|$, primarily arises from the thermal excitation of the intrachain spin degrees of freedom, influenced by both the ferromagnetic intrachain and frustrated interchain interactions. Consequently, the experimental observations of nearly zero specific heat values below ~ 4 K and the gapped behaviors (see Fig. R3.1a) support the premise of the effective model Eq. (5), indicating that α - CoV_2O_6 resides in a subspace of states with nearly identical spins along each chain at $T \leq 4$ K in the low- Γ limit.

At low temperatures, our classical Monte Carlo simulations using the original 3D Hamiltonian with a

large system size (28,800 spins) exhibit a strong tendency for all spins along each chain ($N_b = 50$) to align in the same direction (all-up or all-down) at different longitudinal magnetic fields (Supplementary Fig. 7). Additionally, the integral-path Monte Carlo simulations show that all “spins” along each Trotter chain ($N_b = 50$) tend to take the same state with an average magnetization of each chain $\langle S_I^z \rangle = \frac{1}{N_b} \sum_{I' \in I} \langle S_{I'}^z \rangle = 0.5$ or -0.5 , even at the highest transverse field of $T = 0.1$ K (see Fig. 4 in main text and Supplementary Fig. 12). Below ~ 4 K, the effective model described by Eq. (5) accurately captures both the quasi-equilibrium (see Fig. R3.2b-d) and dynamic (see Supplementary Fig. 8d) properties.

The previous misleading representations of the effective model have been corrected in the revised manuscript. Understanding the complex magnetism of real materials is a challenging task. We would like to emphasize the importance of our experimental findings, which clearly demonstrate quantum annealing effects induced by a small transverse field in the frustrated magnet without evident interaction randomness, for the first time. Furthermore, the effective model described in Eq. (5) provides a good explanation for the experimental results below ~ 4 K. We hope that the Reviewer finds our response and the revised manuscript satisfactory.

Below, I list additional comments for the consideration by the authors:

1) In the simulation shown in Fig. 4, why not use the same initial condition and follow the same protocol as in the experiment?

Reply to Comment 3.3: Following the Reviewer’s suggestion, we have updated the simulation protocol to match the experimental conditions. The simulations now start with the fully-inversely-polarized state at $\mu_0 H^z = -4.2$ T. We gradually raise $\mu_0 H^z$ by 0.1 T at each Monte Carlo step (MCS) until reaching the target field, $\mu_0 dH^z/dt = 10$ mT/s ~ 0.1 T/MCS, and then simulate the relaxation process in subsequent MC steps. We have found no essential difference compared to the previous simulations using the random initial state, and the figures (see the updated Fig. 4 in main text and Supplementary Fig. 12) have been updated accordingly in the revised version. Notably, the updated Fig. 4a,b in the main text reveals more indications of local stripe antiferromagnetic order in the metastable phases at $\mu_0 H^z = 2$ T. This is because, in the updated protocol, the spin system transitions directly from the stripe antiferromagnetic state at $|\mu_0 H^z| \leq 1.5$ T (assuming it to be in equilibrium).

2) On p. 11 of the SI, mI is not an “order parameter” as it is always nonzero for nonzero transverse field. For the same reason, the system does not exhibit a spontaneous transverse magnetization along the x-axis. The transverse magnetization is simply induced by the external field.

Reply to Comment 3.4: We apologize again for the previous misleading representation. We have corrected the erroneous statements and symbols in the revised version. The ground state of the 2D Hamiltonian Eq. (3) is equivalent to that of Eq. (2) in the low- T limit (please see Reply to Comment 3.2). In the Sup-

plementary Information, we have calculated the ground-state wavefunction $|\text{GS}\rangle$ of the 2D Hamiltonian [Eq. (3)] using the finite-size exact diagonalization method. We have further calculated the sublattice magnetization $m^z = \sum_I |\langle \text{GS} | S_I^z | \text{GS} \rangle| / (N_{ac} S)$ and the transverse magnetization $m^x = \sum_I \langle \text{GS} | S_I^x | \text{GS} \rangle / (N_{ac} S)$, which are shown in the revised Supplementary Fig. 10. Here, N_{ac} represents the number of spin chains. These calculations enable us to estimate the minimal strength of the transverse field required to restore symmetry (Γ_c) at 0 K. After carefully reading this comment, we have thoroughly examined our data. We have confirmed that the transverse magnetization m^x is indeed nonzero, albeit very small, and increases as Γ increases, even within the range of $0 < \Gamma < \Gamma_c$. We sincerely appreciate the Reviewer for bringing this error to our attention. We agree that m^x is simply induced by the applied transverse field, and remains nonzero for nonzero transverse field.

List of changes (highlighted in blue in the revised manuscript and supplementary information)

1. The word “apparent” is removed from the abstract and other relevant sections, following the suggestions of Reviewer #1.
2. The discussion on restoring symmetry by a transverse field is incorporated, along with a citation to the new reference, in the third paragraph of the section “Results: Effective spin Hamiltonian”, as per Reviewer #1’s comments.
3. An explanation for why quantum annealing is most clearly observed at 2 T is added in the fourth paragraph of the section “Results: Superiority of quantum annealing over thermal annealing”, addressing the comments of Reviewer #2.
4. The observed classical-quantum crossover behavior is presented as a new section in Supplementary Note 6, including the MPMS data (Supplementary Fig. 17) at the end of the Supplementary Information. It is briefly discussed at the end of the section “Results: Superiority of quantum annealing over thermal annealing” in the main text, as per the comments of Reviewer #2.
5. Rectifications regarding the explanation of domain wall dynamics are made at the end of the section “Results: Topological configurations of metastable states”, addressing the comments of Reviewer #2.
6. A new paragraph discussing the temporal variation of thermal conductivity over a long time scale is added at the end of the section “Results: Heat transport measurements”, and an extended discussion is included at the end of Supplementary Note 5, addressing the comments of Reviewer #2.
7. The data of $H^x \sim 1.7H^z$ are moved from Fig. 2h,i to Supplementary Fig. 17a,b, based on the comments of Reviewer #2.
8. The distribution of the relaxation time (stretching exponents) is discussed in the second paragraph of the section “Results: Superiority of quantum annealing over thermal annealing”, as per the comments of Reviewer #2.
9. Detailed corrections on the effective model are made in the section “Results: Effective spin Hamiltonian” of the main text and Supplementary Note 4, including new data from Supplementary Figs. 9 and 11, addressing the comments of Reviewer #3. In Fig. 1f, “temperature (T)” is replaced with “rescaled temperature (T'')”.
10. The data in Fig. 4 of the main text and Supplementary Fig. 12 are updated to match the experimental conditions, as per the comments of Reviewer #3.
11. Erroneous statements and symbols are corrected in Supplementary Note 4 and Supplementary Fig. 10, addressing the comments of Reviewer #3.

Reviewers' Comments:

Reviewer #1:

Remarks to the Author:

I consider this experimental demonstration of very faster quantum annealing due to transverse field (compared to the earlier results of Science, 1999, by the Group of Aeppli & Rosenbaum in LiHoYF systems) noteworthy.

Because of some very specific modelling for this sample here was developed earlier by the group and extensive numerical results capture the main essence of the observed phenomena here, the reported results will be of significance.

The methodology employed are sound the presentation in this revised version (taking care of all the comments made earlier) is very good.

I recommend its publication in Nature Communication.

Bikas K. Chakrabarti

Reviewer #2:

Remarks to the Author:

The authors made substantial effort to perform additional experiments. Overall, I am satisfied with their response to my review and with the revisions made to the paper. I believe that the revised paper is acceptable for publication in Nature Communications. One thing that should be considered before publication is that in Supplementary Fig. 17, a peak formation is observed in the temperature dependence, but its implications are not describe. I think that the authors should briefly comment on the implications of the peak formation.

Reviewer #3:

Remarks to the Author:

I thank the authors for explaining about their derivation of the effective models in detail. I understand that their argument of equating the number spins per chain in the 3D model with the Trotter number in the effective 2D model, though I would say this is a bit ad hoc, can resolve one issue that I pointed out. However, their explanation also made it clear that their treatment of the effective transverse field is incorrect.

The error can be found in Eq. (2) in their reply (also in the main text with the same equation number), where they argue that the effective transverse field can be obtained by multiplying the real transverse field by the chain length N_b . This is incorrect because the evaluation of the effective transverse field, i.e., the quantum-mechanical tunneling term, requires a perturbative argument. In their notation, the eigenstate S_I^z (say, Up and Down) represents "majority of Co^{2+} ions (magnetic moment) in the I -th chain". This means $|\text{Up}\rangle = |\text{up, up, up, ..., up}\rangle$ and $|\text{Down}\rangle = |\text{down, down, down, ..., down}\rangle$ in terms of the original spin states along a given chain. Hence, a nonzero effective transverse field term can appear at the N_b -th order of the original transverse field, or higher, and it cannot be linear as they argue in Eq. (2).

While I acknowledge that their experimental work is of high quality and quite interesting, due to the error in the theory part, I cannot recommend publication.

Reviewer #1 (Remarks to the Author):

I consider this experimental demonstration of very faster quantum annealing due to transverse field (compared to the earlier results of Science, 1999, by the Group of Aeppli & Rosenbaum in LiHoYF systems) noteworthy.

Because of some very specific modelling for this sample here was developed earlier by the group and extensive numerical results capture the main essence of the observed phenomena here, the reported results will be of significance.

The methodology employed are sound the presentation in this revised version (taking care of all the comments made earlier) is very good.

I recommend its publication in Nature Communication.

Bikas K. Chakrabarti

Reply to Comment 1.1: We thank the Reviewer for his positive assessment and recommendation for publication.

Reviewer #2 (Remarks to the Author):

The authors made substantial effort to perform additional experiments. Overall, I am satisfied with their response to my review and with the revisions made to the paper. I believe that the revised paper is acceptable for publication in Nature Communications. One thing that should be considered before publication is that in Supplementary Fig. 17, a peak formation is observed in the temperature dependence, but its implications are not describe. I think that the authors should briefly comment on the implications of the peak formation.

Reply to Comment 2.1: We thank the reviewer for his/her satisfaction with our previous response and the recommendation for acceptance. The temperature-dependent relaxation rate peaks at $T \sim 3\text{-}4\text{ K}$ ($1/T \sim 0.27\text{-}0.34\text{ K}^{-1}$), indicating a maximum in spin fluctuations. This is likely attributed to a crossover in spin dynamics driven by thermal fluctuations with rising temperature. We have briefly discussed this in the revised manuscript.

Reviewer #3 (Remarks to the Author):

I thank the authors for explaining about their derivation of the effective models in detail. I understand that their argument of equating the number spins per chain in the 3D model with the Trotter number in the effective 2D model, though I would say this is a bit ad hoc, can resolve one issue that I pointed out. However, their explanation also made it clear that their treatment of the effective transverse field is incorrect.

The error can be found in Eq. (2) in their reply (also in the main text with the same equation number), where they argue that the effective transverse field can be obtained by multiplying the real transverse field by the chain length N_b . This is incorrect because the evaluation of the effective transverse field, i.e., the quantum-mechanical tunneling term, requires a perturbative argument. In their notation, the eigenstate S_{Iz} (say, Up and Down) represents majority of Co^{2+} ions (magnetic moment) in the I -th chain. This means $|\text{Up}\rangle = |\text{up, up, up, ..., up}\rangle$ and $|\text{Down}\rangle = |\text{down, down, down, ..., down}\rangle$ in terms of the original spin states along a given chain. Hence, a nonzero effective transverse field term can appear at the N_b -th order of the original transverse field, or higher, and it cannot be linear as they argue in Eq. (2).

While I acknowledge that their experimental work is of high quality and quite interesting, due to the error in the theory part, I cannot recommend publication.

Reply to Comment 3.1: We appreciate the reviewer's recognition of the high quality and interest in our experimental work, as well as his/her professional comments. It is true that the previous Eq. (2) is only exact when $N_b = 1$. Indeed, $\langle \uparrow \dots \uparrow | S_i^z | \downarrow \dots \downarrow \rangle = 0$ with $N_b \geq 2$, rendering the previous theoretical explanation relying on that equation unconvincing. Our experimental findings in $\alpha\text{-CoV}_2\text{O}_6$ clearly demonstrate the occurrence of quantum annealing at low temperatures, even in a tiny transverse field (as low as ~ 3.5 mK). To better understand this phenomenon, it may be necessary to abandon the 2D approximation. Below, we present an updated theoretical explanation directly based on the original 3D Hamiltonian of $\alpha\text{-CoV}_2\text{O}_6$, employing the standard perturbative Monte Carlo (pMC) and stochastic series expansion (SSE) methods.

The original 3D Hamiltonian is formulated with the transverse field per spin (Γ),

$$\mathcal{H} = J_0 \sum_{\langle i, i_0 \rangle} S_i^z S_{i_0}^z + J_1 \sum_{\langle i, i_1 \rangle} S_i^z S_{i_1}^z + J_2 \sum_{\langle i, i_2 \rangle} S_i^z S_{i_2}^z + J_3 \sum_{\langle i, i_3 \rangle} S_i^z S_{i_3}^z - \mu_0 \mu_B H^z g^z \sum_i S_i^z - \Gamma \sum_i S_i^x. \quad (1)$$

We closely follow the pMC method as described in Phys. Rev. B 38, 4712 (1988) and Phys. Rev. B 78, 184408 (2008). We rewrite the Hamiltonian of Eq. (1) as $\mathcal{H} = \mathcal{H}_0 + \mathcal{H}_{\text{TF}}$, where \mathcal{H}_0 represents the classical part and \mathcal{H}_{TF} represents the quantum transverse-field term that does not commute with \mathcal{H}_0 . In the limit of $\beta' \Gamma^2 / |J_0| \ll 1$, where $\beta' \equiv 1/(k_B T)$, we can derive an effective classical Hamiltonian $\mathcal{H}_{\text{eff}}(\psi)$ as a function

of the state $|\psi\rangle$, such that $e^{-\beta'\mathcal{H}_{\text{eff}}(\psi)} = \langle\psi|e^{-\beta'\mathcal{H}}|\psi\rangle$. This effective Hamiltonian is formulated as

$$\mathcal{H}_{\text{eff}} = \mathcal{H}_0 + \beta'\Gamma^2 \sum_i [2S_i^z F_1(\beta'h_i) - F_0(\beta'h_i)], \quad (2)$$

where $h_i = \mu_0\mu_B H^z g^z - J_0 \sum_{\langle i_0 \rangle} S_{i_0}^z - J_1 \sum_{\langle i_1 \rangle} S_{i_1}^z - J_2 \sum_{\langle i_2 \rangle} S_{i_2}^z - J_3 \sum_{\langle i_3 \rangle} S_{i_3}^z$ is the local field at the i th site. The functions $F_0(x)$ and $F_1(x)$ are defined as

$$F_0(x) \equiv \frac{\cosh(x) - 1}{x^2}, F_1(x) \equiv \frac{\sinh(x) - x}{x^2}. \quad (3)$$

The linear transverse-field term indeed disappears as part of a cumulant expansion [Phys. Rev. B 38, 4712 (1988); Phys. Rev. B 78, 184408 (2008)], as anticipated by the reviewer. However, the second (lowest) order term in Γ exists when the same spin is flipped twice within the n th cumulant ($n \geq 2$) [Phys. Rev. B 78, 184408 (2008)].

Qualitatively, at low temperatures (i.e., $x_i = \beta'h_i$ becomes significant), flipping the i th spin along the ferromagnetic chain (see Fig. RR. 1) increases the energy by $\Delta E_{\text{flip}} = \Delta E_0 + \Delta E_{\text{TF}} \sim 2S_i^z h_i - 4S_i^z \beta'\Gamma^2 \sinh(x_i)/x_i^2$, in accordance with the effective classical Hamiltonian of Eq. (2). Here, S_i^z represents the eigenvalue of the i th spin before the flip. At low temperatures (e.g., $T \leq 2$ K), the transverse-field contribution (negative) $\Delta E_{\text{TF}} \sim -4S_i^z \beta'\Gamma^2 \sinh(x_i)/x_i^2$ becomes comparable to the Ising part (positive) $\Delta E_0 = 2S_i^z h_i$. Consequently, the acceptance probability (P) for the flip update (Fig. RR. 1) significantly increases in the Monte Carlo simulation, given by $P = \exp(-\beta'\Delta E_{\text{flip}})$. The weak transverse field effectively weakens the ferromagnetic correlations along the chain (i.e., the b axis) [also see Phys. Rev. B 38, 4712 (1988)], thereby inducing quantum annealing at low temperatures. In contrast, at slightly higher temperatures (e.g. $T \geq 4$ K), ΔE_{TF} becomes negligible compared to ΔE_0 , and the transverse field has minimal impact on the annealing. Therefore, the perturbative theory naturally explains the observed quantum annealing in α -CoV₂O₆ at low temperatures ($T < 4$ K) in the presence of a weak transverse field ($\Gamma \sim 3.5$ mK).

We conducted standard pMC simulations using the effective classical Hamiltonian of Eq. (2) at $T \geq 1.8$ K and $\Gamma \leq 7$ mK. With $\beta'\Gamma^2/|J_0| \leq 9 \times 10^{-7} \ll 1$ and $\beta'\Gamma \leq 0.004 \ll 1$, the perturbative theory remains valid. The successful simulations of the quasi-equilibrium-state thermodynamic properties of α -CoV₂O₆ (see Fig. RR. 2) provide further validation for this perturbative approach. However, for larger transverse fields or lower temperatures, the pMC method may result in unphysical ground states, making it unreliable [Phys. Rev. B 38, 4712 (1988); Phys. Rev. B 78, 184408 (2008)].

We further simulated the annealing process of α -CoV₂O₆ at $T \geq 1.8$ K and $\Gamma \leq 7$ mK, as depicted in Fig. RR. 3. These simulations were performed on a $12 \times 12 \times 24$ (3,456-site) cluster with periodic boundary conditions, with the longest dimension aligned along the chain (i.e., b axis). Precisely calculating ΔE_{TF} ,

Fig. RR1 | Diagram of the single-spin-flip update along the ferromagnetic chain.

Fig. RR2 | Simulations of quasi-equilibrium-state thermodynamic properties for α - CoV_2O_6 at various small transverse fields (Γ). **a** magnetic specific heat, **b** susceptibility, and **c** magnetization calculated using the original 3D Hamiltonian of Eq. (1). Methods used include classical Monte Carlo (CMC) at $\Gamma = 0$ K, perturbative Monte Carlo (pMC) at $\Gamma = 7$ mK, and stochastic series expansion (SSE) quantum Monte Carlo at $\Gamma = 0.1$ K. Experimental data are presented as scatter plots for comparison. The same set of refined interaction parameters (as described in the manuscript) is used. The weak Γ has little impact on these quasi-equilibrium-state thermodynamic properties, in excellent agreement with experimental results (refer to Supplementary Fig. 9).

which involves contributions from spins surrounding the flipping site [as detailed in Eq. (2)], entails higher computational costs. Notably, our simulations demonstrate that even a tiny transverse field of $\Gamma \sim 3.5$ mK can significantly reduce the relaxation time at $T \sim 1.8$ K after enough MCS (see Fig. RR. 3a,b), qualitatively consistent with our experimental findings. Furthermore, it is noteworthy that a few (random) spin chains along the b axis do not exhibit perfect ferromagnetic ordering in out-of-equilibrium states when a weak transverse field is present (with $\mu_0 H^z = 0$ T, as shown in Fig. RR. 3c compared to Fig. RR. 3d. A similar conclusion can be drawn for $\mu_0 H^z = 2$ T, as shown in Fig.4h of the revised main text), in line with the aforementioned qualitative analysis. At low temperatures, the weak transverse field can effectively weaken the ferromagnetic correlations along the chain [Phys. Rev. B 38, 4712 (1988)], thereby expediting the

Fig. RR3 | Perturbative Monte Carlo simulations of relaxation processes for α -CoV₂O₆ under various transverse fields (Γ). **a** Monte Carlo step (MCS) dependence of the residual energy per site ($E - E_{GS}$) calculated at $T = 1.8$ K and $\mu_0 H^z = 0$ T. **b** MCS dependence of the magnetization (M^z) calculated at $T = 1.8$ K and $\mu_0 H^z = 2$ T. Both datasets are averaged over 50 independent samples. Out-of-equilibrium spin configurations along the ferromagnetic chain (i.e., b axis) after 200,000 MCS at $T = 1.8$ K and $\mu_0 H^z = 0$ T, with $\Gamma = 7$ mK (**c**) and 0 K (**d**). All the chains with deviations from perfect ferromagnetic order are highlighted. The simulations started with the fully-inversely-polarized state at $\mu_0 H^z = -4.2$ T, and were performed on a $12 \times 12 \times 24$ (3,456-site) cluster with periodic boundary conditions and with the longest dimension along the chain. To follow the experimental protocol, we gradually raised $\mu_0 H^z$ by 0.1 T at each MCS until reaching the target field and then simulated the relaxation process in subsequent MCS.

annealing process towards the ground state. In sharp contrast, at $\Gamma \sim 0$ K, nearly all the spin chains display perfect ferromagnetic order after only ~ 400 MCS at low temperatures. Therefore, in subsequent MCS a substantial energy barrier is necessary to flip the perfectly ferromagnetic chains to reach a lower-energy state at $\Gamma \sim 0$ K, owing to the frustrated interchain interactions.

For larger transverse fields and lower temperatures, we conducted stochastic series expansion (SSE) quantum Monte Carlo simulations [Phys. Rev. E 68, 056701 (2003); Phys. Rev. B 103, 104416 (2021)] on a $6 \times 6 \times 12$ (432-site) cluster with periodic boundary conditions. The longest dimension is along the b axis. The successful simulations of the quasi-equilibrium-state thermodynamic properties at $\Gamma = 0.1$ K indicate negligible finite-size effects and validate this approach (see Fig. RR. 2). We also simulated the annealing

Fig. RR4 | Quantum Monte Carlo simulations for α -CoV₂O₆ using the SSE method. **a, b** Monte Carlo step (MCS) dependence of the energy per site (E) calculated at $T = 1$ K in $\mu_0 H^z = 0$ and 2 T. Both datasets are averaged over 64 independent samples. The simulations started with random states at $\mu_0 H^z = -4.2$ T^a. We gradually raised $\mu_0 H^z$ until reaching the target field and then simulated the relaxation process in subsequent MCS. **c** Transverse field dependence of staggered magnetization $m_s^z = \langle |\sum_i S_i^z (-1)^{i_a+i_c}| \rangle / (NS)$, where N is the site number, $S = 1/2$, and i_a and i_c are site indexes along the a and c axes, respectively. The critical transverse field $\Gamma_c \sim 36$ K is marked. All the calculations were performed on a $6 \times 6 \times 12$ (432-site) cluster with periodic boundary conditions.

^a Due to frustration combined with a low transverse field or high longitudinal field, the SSE Wolff cluster algorithm is less efficient at low temperatures. To make the relaxation processes clear, we initiated all the simulations with random states at $\mu_0 H^z = -4.2$ T instead of the fully-inverse-polarized state as used in pMC. Classical Monte Carlo simulations indicate that the spin system should quickly relax to the fully-inversely-polarized state after only 3-4 MCS at $\mu_0 H^z \sim -4$ T, please refer to Supplementary Fig. 7h.

processes of α -CoV₂O₆ at $T = 1$ K in $\mu_0 H^z = 0$ and 2 T under various transverse fields (see Fig. RR. 4a, b). These SSE simulations also clearly demonstrate that the weak transverse field ($\Gamma \leq 0.1$ K) has minimal impact on the ground state and quasi-equilibrium-state thermodynamic properties (Fig. RR. 2), but can effectively expedite the annealing process towards the ground state at low temperatures. Furthermore, at low temperatures, the critical transverse field that completely suppresses the classical order is calculated to be $\Gamma_c \sim 36$ K (comparable to $|J_0|$), and the reduction of the staggered magnetization m_s^z at $\Gamma < 0.1$ K is estimated to be less than $\sim 1 \times 10^{-5}$ (i.e., $\sim 1 \times 10^{-4} \mu_B/\text{Co}$) (see Fig. RR. 4c). Based on these SSE results using the original 3D Hamiltonian of α -CoV₂O₆ [Eq. (1)], we now recognize that the role of the transverse field was significantly overestimated in the previous 2D approximation.

We apologize for the error in our previous theoretical explanation and thank the reviewer for bringing it to our attention. Understanding the many-body-correlated quantum magnetism of a real material is extremely challenging. We have conducted a comprehensive experimental investigation on high-quality single crystals of α -CoV₂O₆, which clearly demonstrates quantum annealing induced by a tiny transverse field at low temperatures. In the revised manuscript, we have abandoned the 2D approximation and, instead, rein-

terpreted the experimental results based on the original 3D Hamiltonian, employing both the standard pMC and SSE methods. We hope the reviewer finds the updated theoretical explanation satisfactory.

List of changes (highlighted in blue in the revised manuscript and supplementary information)

1. We briefly discuss the implications of the peak formation in the last paragraph of the section titled “Results: Superiority of quantum annealing over thermal annealing”, in response to the suggestions of Reviewer #2.
2. All the theoretical explanations based on the incorrect reduced 2D Hamiltonian have been completely removed from both the manuscript and supplementary information. Instead, we have reinterpreted the experimental observations based on the original 3D Hamiltonian using the standard perturbative Monte Carlo and stochastic series expansion methods, in accordance with the comments from Reviewer #3.
3. The figures in the main text, specifically Figs. 1 and 4, have been adjusted accordingly to reflect the updated theoretical explanation. The figures in the supplementary information have also been updated to align with these changes.
4. Both the perturbative Monte Carlo and stochastic series expansion methods are extensively discussed in the revised Supplementary Note 4.

Reviewers' Comments:

Reviewer #3:

Remarks to the Author:

Throughout the earlier rounds, I have been questioning about their theoretical interpretation using MC methods by pointing out technical, but critical, issues, even though I acknowledge the high quality and the high value of their experimental work. In the latest manuscript, the authors retracted their almost entire previous quantum MC simulation results and replaced them with new results using the pMC method and the SSE method. In my opinion, however, the new results are still not enough and have potential errors to give a reasonable theoretical support for their experimental findings for the following reasons:

1) Possible misinterpretation of the effective Hamiltonian used in the pMC method

2) Failure to present theoretical demonstration of quantum annealing effects using QMC simulations, contrary to the claim by the authors

Therefore, unfortunately, I disagree with the opinions of the other referees and do not support the publication of the current manuscript in nature communication.

In the following, I will explain more details about these issues.

1) The effective Hamiltonian H_{eff} (2) is derived from a second-order perturbation theory applied to the density matrix using a standard cumulant expansion. H_{eff} includes the F1 term arising from quantum fluctuations, which is an analog of a diamagnetic field (F1) term reducing the classical molecular field acting on each site. Here, since the F1 term grows exponentially at low temperatures, the interpretation requires more cautions than the arguments given by the authors. Particularly, it can be a misinterpretation of the perturbation theory to believe that the F1 term can be comparable to the zeroth order term.

To see this point, one can first consider a single-site problem with the Hamiltonian $H = -h S_z - \Gamma S_x$, where h may be regarded as a molecular field in a sense of a cluster mean field theory. On one hand, one can easily evaluate the exact density matrix to derive $W_{\uparrow} = \langle \uparrow | \exp(-\beta H) | \uparrow \rangle$ and $W_{\downarrow} = \langle \downarrow | \exp(-\beta H) | \downarrow \rangle$. On the other hand, one can use the effective Hamiltonian to evaluate $W_{\uparrow, \text{eff}} = \langle \uparrow | \exp(-\beta H_{\text{eff}}) | \uparrow \rangle$ and $W_{\downarrow, \text{eff}} = \langle \downarrow | \exp(-\beta H_{\text{eff}}) | \downarrow \rangle$. With the two results, one can compare the weight ratios $W_{\downarrow} / W_{\uparrow}$ and $W_{\downarrow, \text{eff}} / W_{\uparrow, \text{eff}}$ in the exact and the effective treatments. Here, we may consider $\beta h \sim |J_0| / 1.8\text{K} \sim 17$ and $\Gamma / h \sim \Gamma / |J_0| \sim 2.3 \times 10^{-4}$ by referring to the experiments. One can show that while indeed $W_{\downarrow, \text{eff}} / W_{\uparrow, \text{eff}}$ increases at low temperatures as the authors argue, the trend considerably overestimates that of the exact result.

With that said, I cannot help but wonder why the authors had to invoke the approximate simulation (pMC) given that they have a capability to carry out SSE. While the latter may require more computational resource, but the amount of the required task seems within a standard level. In fact, the authors have carried out SSE simulations for a larger transverse field. The simulation needed here for a smaller transverse field is less computationally expensive.

2)

In the end, what numerical/theoretical evidence has been provided by the authors? They have shown the decrease of the energy, but this is simply expected for any second order perturbation theory (Fig. 4i). They have shown some snapshot of vortices-antivortices, but with the lack of quantitative analysis, it is hard to see to what degrees these observations are relevant in relation with the experiments (Fig. 4a-d). With the possible issue I mentioned above in 1) put aside for now, the pMC simulation yields snapshots where a few spins are flipped here and there along each chain, but this seems quite different from what is expected for quantum-mechanical tunneling events between different ground states (Fig. 4g, h). Finally, with the possible issue with the pMC put aside once again, the trend of a slightly faster relaxation the magnetization for $\Gamma = 7\text{mK}$ seems

taken over at a later time by the cases with smaller transverse fields (Fig. 4j), leaving a confusion that this may or may not be related to the experiments.

Reviewer #3 (Remarks to the Author):

Comment 1: Throughout the earlier rounds, I have been questioning about their theoretical interpretation using MC methods by pointing out technical, but critical, issues, even though I acknowledge the high quality and the high value of their experimental work. In the latest manuscript, the authors retracted their almost entire previous quantum MC simulation results and replaced them with new results using the pMC method and the SSE method. In my opinion, however, the new results are still not enough and have potential errors to give a reasonable theoretical support for their experimental findings for the following reasons:

- 1) Possible misinterpretation of the effective Hamiltonian used in the pMC method
- 2) Failure to present theoretical demonstration of quantum annealing effects using QMC simulations, contrary to the claim by the authors

Therefore, unfortunately, I disagree with the opinions of the other referees and do not support the publication of the current manuscript in nature communication.

In the following, I will explain more details about these issues.

1) The effective Hamiltonian \mathcal{H}_{eff} (2) is derived from a second-order perturbation theory applied to the density matrix using a standard cumulant expansion. \mathcal{H}_{eff} includes the F1 term arising from quantum fluctuations, which is an analog of a diamagnetic field (F1) term reducing the classical molecular field acting on each site. Here, since the F1 term grows exponentially at low temperatures, the interpretation requires more cautions than the arguments given by the authors. Particularly, it can be a misinterpretation of the perturbation theory to believe that the F1 term can be comparable to the zeroth order term.

To see this point, one can first consider a single-site problem with the Hamiltonian $\mathcal{H} = -hS^z - \Gamma S^x$, where h may be regarded as a molecular field in a sense of a cluster mean field theory. On one hand, one can easily evaluate the exact density matrix to derive $W_{\uparrow} = \langle \uparrow | \exp(-\beta\mathcal{H}) | \uparrow \rangle$ and $W_{\downarrow} = \langle \downarrow | \exp(-\beta\mathcal{H}) | \downarrow \rangle$. On the other hand, one can use the effective Hamiltonian to evaluate $W_{\uparrow,\text{eff}} = \langle \uparrow | \exp(-\beta\mathcal{H}_{\text{eff}}) | \uparrow \rangle$ and $W_{\downarrow,\text{eff}} = \langle \downarrow | \exp(-\beta\mathcal{H}_{\text{eff}}) | \downarrow \rangle$. With the two results, one can compare the weight ratios $W_{\downarrow}/W_{\uparrow}$ and $W_{\downarrow,\text{eff}}/W_{\uparrow,\text{eff}}$ in the exact and the effective treatments. Here, we may consider $\beta h \sim |J_0|/1.8 \text{ K} \sim 17$ and $\Gamma/h \sim \Gamma/|J_0| \sim 2.3 \times 10^{-4}$ by referring to the experiments. One can show that while indeed $W_{\downarrow,\text{eff}}/W_{\uparrow,\text{eff}}$ increases at low temperatures as the authors argue, the trend considerably overestimates that of the exact result.

With that said, I cannot help but wonder why the authors had to invoke the approximate simulation (pMC) given that they have a capability to carry out SSE. While the latter may require more computational resource, but the amount of the required task seems within a standard level. In fact, the authors have carried out SSE simulations for a larger transverse field. The simulation needed here for a smaller transverse field is less computationally expensive.

Reply 1: We sincerely thank the Reviewer for recognizing the high quality and value of our experimental

work, as well as for providing helpful comments on the theoretical interpretation of the quantum annealing effects induced by a small transverse field ($\Gamma \leq 0.1$ K). Below, we respond to the specific and detailed comments raised by the Reviewer one by one.

After careful re-examination, we affirm that the perturbative Monte Carlo (pMC) method remains largely valid for $T \geq 1.8$ K and $\Gamma \leq 7$ mK. However, recognizing potential limitations in the pMC method, particularly at low temperatures, we have substantially revised the manuscript. The focus is now only on the stochastic series expansion (SSE) quantum Monte Carlo (QMC) simulations, which have been incorporated into the main text.

Firstly, to validate the pMC method, we examine calculated equilibrium-state thermodynamic properties as a function of temperature, following previous studies [e.g., Phys. Rev. B 78, 184408 (2008)]. Due to the extremely long relaxation times in the low transverse field at low temperatures (~ 1.8 K) using the original spin Hamiltonian of α -CoV₂O₆, it is challenging to compute the fully equilibrium-state thermodynamic properties. Responding to the Reviewer's suggestion, we employed the single-site molecular-field model to calculate the fully equilibrium-state energy, providing a means to validate the perturbation theory. As illustrated in Fig. R1a, the equilibrium-state energies calculated from the effective and exact Hamiltonians perfectly overlap above ~ 1.8 K, at $\Gamma = 7$ mK.

Fig. R1 | Calculations for the single-site molecular-field model. a, b Equilibrium-state energies (E) and density matrix ratios ($W_{\downarrow}/W_{\uparrow}$ and $W_{\downarrow,\text{eff}}/W_{\uparrow,\text{eff}}$) calculated using the exact Hamiltonian $\mathcal{H} = -hS^z - \Gamma S^x$ and the effective Hamiltonian $\mathcal{H}_{\text{eff}} = -hS^z + \beta\Gamma^2(2S^z F_1(\beta h) - F_0(\beta h))/4$, employing a mean field theory. Here, $F_0(x) \equiv \frac{\cosh(x)-1}{x^2}$ and $F_1(x) \equiv \frac{\sinh(x)-x}{x^2}$. Inset of b displays $W_{\downarrow,\text{eff}}W_{\uparrow}/W_{\uparrow,\text{eff}}W_{\downarrow}$.

Secondly, in response to the Reviewer's suggestion, we further calculated the exact and perturbative/effective density elements based on the same molecular-field model. The spin-up exact density element

is computed as $W_{\uparrow} = \langle \uparrow | \exp(-\beta \mathcal{H}) | \uparrow \rangle = \sum_{j_1=1,2} \sum_{j_2=1,2} \langle \uparrow | \psi_{j_1} \rangle \langle \psi_{j_1} | \exp(-\beta \mathcal{H}) | \psi_{j_2} \rangle \langle \psi_{j_2} | \uparrow \rangle$, where $\langle \psi_{j_1} | \exp(-\beta \mathcal{H}) | \psi_{j_2} \rangle = \delta_{j_1, j_2} \exp(-\beta E_{j_1})$, and $E_1, E_2, |\psi_1\rangle, |\psi_2\rangle$ are the eigenenergies and eigenstates of $\mathcal{H} = -hS^z - \Gamma S^x$. Similarly, we obtain W_{\downarrow} . On the other hand, the density element can be expressed in terms of a cumulant expansion, $W_{\uparrow} = \exp[-\beta \langle \uparrow | \mathcal{H} | \uparrow \rangle + \sum_{n=2}^{\infty} \frac{(-\beta)^n}{n!} \langle \uparrow | (\mathcal{H} - \langle \uparrow | \mathcal{H} | \uparrow \rangle)^n | \uparrow \rangle]$ [Phys. Rev. B 78, 184408 (2008)]. Typically, one keeps to the lowest order $O(\Gamma^2)$, $\langle \uparrow | (\mathcal{H} - \langle \uparrow | \mathcal{H} | \uparrow \rangle)^n | \uparrow \rangle \sim \Gamma^2 h^{n-2}/4$, and obtains the effective density element $W_{\uparrow, \text{eff}} = \exp[\beta h/2 - \beta^2 \Gamma^2 [F_1(\beta h) - F_0(\beta h)]/4] = \langle \uparrow | \exp(-\beta \mathcal{H}_{\text{eff}}) | \uparrow \rangle$. Here, $\mathcal{H}_{\text{eff}} = -hS^z + \beta \Gamma^2 [2S^z F_1(\beta h) - F_0(\beta h)]/4$ with $F_0(x) \equiv \frac{\cosh(x)-1}{x^2}$ and $F_1(x) \equiv \frac{\sinh(x)-x}{x^2}$. Similarly, we obtain $W_{\downarrow, \text{eff}} = \exp[-\beta h/2 + \beta^2 \Gamma^2 [F_1(\beta h) + F_0(\beta h)]/4]$. The weight ratios calculated for the exact ($W_{\downarrow}/W_{\uparrow}$) and effective ($W_{\downarrow, \text{eff}}/W_{\uparrow, \text{eff}}$) models are shown in Fig. R1b. Clearly, both the exact $W_{\downarrow}/W_{\uparrow}$ and effective $W_{\downarrow, \text{eff}}/W_{\uparrow, \text{eff}}$ are profoundly enhanced by a tiny transverse field Γ , suggesting the quantum tunneling effect of Γ at 1.8 K. Moreover, the difference between $W_{\downarrow}/W_{\uparrow}$ and $W_{\downarrow, \text{eff}}/W_{\uparrow, \text{eff}}$ is not significant at $\Gamma \leq 7$ mK, with $1 \leq W_{\downarrow, \text{eff}} W_{\uparrow} / W_{\uparrow, \text{eff}} W_{\downarrow} \leq 1.047$ at 1.8 K (see inset of Fig. R1b), thereby supporting the validation of the pMC method. At $\Gamma = 7$ mK and $T = 1.8$ K, the magnetization decreases caused by Γ (and T) are calculated, $\Delta m_{\Gamma} = 1 - \langle m \rangle = 1.03 \times 10^{-7}$ and $\Delta m_{\Gamma, \text{eff}} = 1.08 \times 10^{-7}$, further supporting the validation of the pMC method. $\Delta M_{\Gamma}^z \sim \Delta M_{\Gamma, \text{eff}}^z \sim 1 \times 10^{-6} \mu_B/\text{Co}$ (at $\Gamma = 7$ mK and $T = 1.8$ K) suggests that the small transverse field has a negligible effect on the equilibrium-state magnetization (for the equilibrium-state energy ΔE_{Γ} , please refer to our reply to Comment 2), where $\Delta M_{\Gamma}^z = g \Delta m_{\Gamma}/2$ and $\Delta M_{\Gamma, \text{eff}}^z = g \Delta m_{\Gamma, \text{eff}}/2$.

Given that $\Delta E_{\text{flip}} = -k_B T \ln(W_{\downarrow}/W_{\uparrow})$ represents the effective barrier energy for the spin flip, the decrease in ΔE_{flip} induced by the transverse field (Γ) can be calculated as $\Delta E_{\text{TF}} = \Delta E_0 - \Delta E_{\text{flip}}$, where $\Delta E_0 = h$ (~ 30.73 K) is the barrier energy at $\Gamma = 0$ K. Using the same molecular-field model, at $\Gamma = 7$ mK, we obtain $\Delta E_{\text{TF}} = 0.5$ K at $T = 1.8$ K ($\Delta E_{\text{TF}} = 0.5$ and 0.6 K are obtained using the exact and effective Hamiltonians, respectively), and $\Delta E_{\text{TF}} = 12.6$ K at $T = 1$ K (using the exact Hamiltonian, as at 1 K the pMC method becomes seriously unreliable as shown in Fig. R1a). Therefore, the reduce of the effective barrier energy ($\Delta E_{\text{TF}}, \gg \Gamma$) can indeed become comparable to ΔE_0 at low temperatures. We have omitted discussions on ΔE_{TF} and ΔE_{flip} in the revised manuscript to avoid ambiguity, and the quantum tunneling effect of small Γ is already indicated by the increase in $W_{\downarrow}/W_{\uparrow}$.

Finally, in response to the Reviewer's suggestion, we have conducted SSE-QMC simulations for smaller transverse fields (Figs. R2 and R3). Despite distinct differences in the update algorithms and cluster sizes, these SSE results roughly/qualitatively agree with the pMC results at 1.8 K (see Fig. R2). Notably, quantum annealing effects induced by the small transverse field (Γ) are evident in the SSE simulations at 1 K and for $0.02 \leq \Gamma \leq 0.1$ K (see Fig. R3), but become vaguely visible at 1.8 K and $\Gamma < 0.02$ K (Fig. R2, please refer to our reply to Comment 2 for possible explanations).

Fig. R2 | Comparison of pMC and SSE simulations at 1.8 K. **a, b** Monte Carlo step (MCS) dependence of the energy per site calculated at zero longitudinal magnetic field ($\mu_0 H^z = 0$ T), E_{pMC} and E_{SSE} . **c, d** MCS dependence of the longitudinal magnetization at $\mu_0 H^z = 2$ T, M_{pMC}^z and M_{SSE}^z . The original spin Hamiltonian of $\alpha\text{-CoV}_2\text{O}_6$ is used. Please note that the classical and quantum update algorithms differ significantly.

Therefore, we affirm the continued validity of the pMC method for $\Gamma \leq 7$ mK and $T \geq 1.8$ K, especially in larger-size simulations, and it may provide complementary support to the SSE-QMC results. Recognizing the potential limitations of pMC at lower temperatures and higher transverse fields, we have heeded the Reviewer's suggestion and shifted the focus to SSE-QMC simulations in the revised main text. Reducing the transverse field markedly decreases the quantum algorithm's efficiency, as it induces slower spin dynamics at low temperatures. This, in turn, escalates the computational cost of SSE computations, necessitating an increase in the number of Monte Carlo steps. In the updated version, we have performed additional SSE-QMC simulations for smaller transverse fields, as advised by the Reviewer. Currently, all SSE-QMC data have been incorporated in Fig. 4, replacing the pMC results, and are extensively discussed in the revised main text.

Comment 2: 2) In the end, what numerical/theoretical evidence has been provided by the authors? They have shown the decrease of the energy, but this is simply expected for any second order perturbation theory (Fig. 4i). They have shown some snapshot of vortices-antivortices, but with the lack of quantitative

analysis, it is hard to see to what degrees these observations are relevant in relation with the experiments (Fig. 4a-d). With the possible issue I mentioned above in 1) put aside for now, the pMC simulation yields snapshots where a few spins are flipped here and there along each chain, but this seems quite different from what is expected for quantum-mechanical tunneling events between different ground states (Fig. 4g, h). Finally, with the possible issue with the pMC put aside once again, the trend of a slightly faster relaxation the magnetization for $\Gamma = 7$ mK seems taken over at a later time by the cases with smaller transverse fields (Fig. 4j), leaving a confusion that this may or may not be related to the experiments.

Fig. R3 | SSE simulations at 1 K. **a** Density matrix ratios, $W_{\downarrow}/W_{\uparrow}$. Here, $W_{\uparrow} = \langle \uparrow | \exp(-\beta\mathcal{H}) | \uparrow \rangle$ and $W_{\downarrow} = \langle \downarrow | \exp(-\beta\mathcal{H}) | \downarrow \rangle$ are exactly calculated using the single-site molecular-field Hamiltonian $\mathcal{H} = -hS^z - \Gamma S^x$. **b** Monte Carlo step (MCS) dependence of longitudinal magnetization calculated at $\mu_0 H^z = 2$ T. **c, d** MCS dependence of the energy per site (E) calculated at $\mu_0 H^z = 0$ and 2 T. The original spin Hamiltonian of α -CoV₂O₆ is used in b-d.

Reply 2: The numerical results primarily indicate that the tiny transverse field ($\Gamma \leq 0.1$ K) doesn't significantly alter the quasi-equilibrium-state thermodynamic properties (please refer to Supplementary Fig. 10), but effectively expedites the annealing process towards the ground state at low temperatures (see Fig. R3), qualitatively consistent with the experimental findings.

In the valid range of the pMC method, $T \geq 1.8$ K and $\Gamma \leq 7$ mK, the equilibrium-state energy change

induced by the transverse field, calculated using the perturbation theory, is entirely negligible. We utilize the single-site molecular-field model proposed by the Reviewer ($\mathcal{H} = -hS^z - \Gamma S^x$, see Comment 1). At $T = 1.8$ K, the energy decrease caused by $\Gamma = 7$ mK can be computed using the effective classical Hamiltonian, $\Delta E_{\Gamma,\text{eff}} = -\beta\Gamma^2[F_1(\beta h) - F_0(\beta h)]/4 \sim 3.8 \times 10^{-7}$ K. The exact decrease in the ground energy caused by $\Gamma = 7$ mK is even slightly larger, $\Delta E_{\Gamma} \sim \sqrt{\Gamma^2 + h^2}/2 - h/2 \sim 4.0 \times 10^{-7}$ K ($\sim \Delta E_{\Gamma,\text{eff}}$). In the limit of $\beta h \rightarrow \infty$ and $\Gamma/h \rightarrow 0$ (with $\Gamma/h \leq 2.3 \times 10^{-4}$ in our case), both $\Delta E_{\Gamma,\text{eff}}$ and ΔE_{Γ} converge to the same value of $\Gamma^2/h/4$ ($\sim 4.0 \times 10^{-7}$ K at $\Gamma = 7$ mK), confirming the validation of the perturbation theory. Clearly, the observed decrease in energy per site caused by the tiny transverse field, > 0.2 K at large Monte Carlo steps (MCS) (see Figs. R2a, b and R3c, d), cannot be attributed to the equilibrium-state energy decrease or the perturbation theory itself. Therefore, the substantial energy decrease caused by the transverse field, significantly larger than $\sqrt{\Gamma^2 + h^2}/2 - h/2$, should be attributed to the many-body quantum annealing effect of Γ . Furthermore, at small MCS, the energies calculated at various transverse fields ($\Gamma \leq 0.1$ K) align with each other in both SSE-QMC and pMC simulations (see Figs. R2a, b and R3c, d). The distinct energy decrease caused by Γ becomes apparent only at larger MCS and lower temperatures (see Fig. R3c, d), further confirming the quantum annealing effect observed in the simulations.

In response to the reviewer's suggestion, we have excluded the snapshots presented in the previous Fig. 4a-d as they lack quantitative analyses relevant to the experiments. In the revised version, we focus on the MCS dependence of energy and longitudinal magnetization. While we initially considered using the snapshots in the previous Fig. 4g, h to illustrate the quantum tunneling capacity of the small transverse field, they appeared distinct from the relaxation processes. Consequently, in this revised version, we have also omitted these snapshots, as the quantum tunneling effects of the small transverse field has already been elucidated in the MCS dependence of energy and longitudinal magnetization calculated using the original Hamiltonian, as well as in $W_{\downarrow}/W_{\uparrow}$ calculated using the simplified model (Fig. R3a).

At $T = 1$ K, the ratios $W_{\downarrow}/W_{\uparrow}$ at $\Gamma = 3.5$ mK and 0.02 K are estimated to be $\sim 3 \times 10^{-9}$ and 1×10^{-7} , respectively, markedly larger than $W_{\downarrow}/W_{\uparrow} \sim 5 \times 10^{-14}$ at $\Gamma = 0$ K (see Fig. R3a). To observe clear quantum annealing effects of Γ , a minimal number of MCS roughly proportional to $\sim W_{\uparrow}/W_{\downarrow}$ is required, resulting in $W_{\uparrow}/W_{\downarrow} \sim 3 \times 10^8$ ($\Gamma = 3.5$ mK) and 9×10^6 ($\Gamma = 0.02$ K). Although $W_{\downarrow}/W_{\uparrow}$ is also significantly enhanced by $\Gamma = 3.5$ mK, much larger MCS are needed to observe clear quantum annealing effects compared to $\Gamma = 0.02$ K. At $\Gamma = 0.02$ K, quantum annealing effects are evident only after $\sim 3 \times 10^4$ SSE MCS ($\times 100$) (see Fig. R3b, d), necessitating more than $\sim 1 \times 10^6$ SSE MCS ($\times 100$) at $\Gamma = 3.5$ mK. Therefore, observing clear quantum annealing effects at $0 < \Gamma < 0.02$ K remains extremely challenging due to the increased computational cost associated with larger MCS.

At 1.8 K, the quantum tunneling capacity of Γ (indicated by the increase in $W_{\downarrow}/W_{\uparrow}$ due to Γ) is signif-

icantly reduced compared to that at 1 K (refer to Fig. R3a). Consequently, the quantum annealing effects of Γ at 1.8 K are expected to be notably weaker. Secondly, to mitigate the variability in the calculated MCS dependence of energy and magnetization, we averaged over ≥ 50 independent samples (despite the associated high computational cost). However, achieving complete elimination of this variability remains challenging in the MC simulations, especially at higher temperatures around 1.8 K, where thermal fluctuations come into play (see Fig. R2). Thirdly, while the present microscopic model is precise enough to account for the quasi-equilibrium-state thermodynamic properties (see Supplementary Fig. 10), it may lack high precision in simulating the slow spin dynamics observed in α -CoV₂O₆, especially at small transverse fields ($0 < \Gamma < 0.02$ K, see Fig. R2). This discrepancy may arise from a low concentration of structural defects and internal transverse magnetic fields contributed by the dipole moments of Co, which are unavoidable in the real material and complexities not considered in our simulations. Finally, the pMC method may lack high reliability or precision at 1.8 K especially in a longitudinal magnetic field of 2 T (Fig. R2c), potentially introducing inconsistencies in the calculated magnetization trends across various transverse fields, at large MCS. We appreciate the Reviewer's helpful comments and thorough examination of the theoretical interpretation. It is important to note that the main focus of this work is to present experimental findings regarding the quantum annealing effects of the transverse field ($\Gamma \leq 0.1$ K) in α -CoV₂O₆. The present numerical results seek to offer a qualitative interpretation of the annealing effects of the tiny transverse field at low temperatures (see Fig. R3). Achieving quantitative agreement with the observed slow spin dynamics in α -CoV₂O₆, particularly at $0 < \Gamma < 0.02$ K, requires further investigations.

We acknowledge that the pMC simulations may become unreliable at low temperatures and high transverse fields. Consequently, in the revised version, we conducted a significantly increased number of SSE computations, extending to larger MCS and averaging over 64 independent samples. Our focus shifted to these SSE results, revealing a more consistent trend across various transverse fields at large MCS at 1 K (see Fig. R3). The aforementioned discussions have been incorporated, and we hope the Reviewer finds our responses and the updated manuscript satisfactory.

List of changes (highlighted in blue in the revised manuscript and supplementary information)

1. We have tempered the statements regarding the many-body simulations throughout the manuscript, explicitly stating that the Monte Carlo analyses for the slow spin dynamics of α -CoV₂O₆ are qualitative. The word “well-defined” has been removed from the title.

2. We have added the extended discussion of the SSE-QMC simulations to the section of “Quantum Monte Carlo simulations” in the main text, and transferred the approximate pMC simulations to Supplementary Note 4.

3. The diagram of the single-spin-flip update in the previous Fig. 1e of the main text has been removed.

4. Fig. 4 in the main text has been revised to exclusively depict the SSE-QMC results. The snapshots of spin configurations have been excluded in response to the Reviewer’s suggestion.

5. The validation of pMC simulation is comprehensively discussed in Supplementary Note 4. A minor error in the effective Hamiltonian has been identified and rectified, leading to the update of the corresponding pMC results. Phys. Rev. B 78, 184408 (2008) utilized Pauli operators, whereas we employed spin operators, introducing an additional factor of “1/2” in front of “ I ” in the effective Hamiltonian.

Reviewers' Comments:

Reviewer #3:

Remarks to the Author:

I thank the authors for responding to my questions and comments. In the latest manuscript, their main theoretical results are now based on the SSE QMC simulation. I would not further comment on their responses about the perturbative MC simulation because the corresponding results have been removed from the manuscript. In any case, I fully appreciate the additional efforts taken by the authors, where they finally decided to take the current standard technique (after many stages of revisions).

With that said, I have to point out that their simulation was merely able to provide a qualitative description of QA effects -- "these simulations unequivocally demonstrate QA effects caused by the small transverse field (...), providing a qualitative description of the previously mentioned experimental observations." The magnitude of the transverse field relevant to their experiment is below 7 mK, whereas the relaxation of the magnetization in this energy scale barely shows any enhancement of the relaxation (Figs. 4f and 4i). Since the relaxation is too slow in simulations and the required magnitude for the transverse field (0.02K?) is at least around 3 times larger (which I think is a rather optimistic estimate) than 7 mK, it seems to me that claiming any connection with their experiments is hard. Although the authors mention structural defects or internal transverse magnetic fields, these conjectures are no more than speculative. Regarding the internal energy mentioned in the new manuscript (Figs. 4d, 4e, 4g, and 4h), as I mentioned in my previous report, the decrease of energy would be found in any second order perturbative process (that is, uncorrelated spin flips back and forth) and I would not consider it as an enough evidence of QA towards a different ground state.

To conclude, my assessment of the manuscript is such that although the experiments might show some interesting QA effects, the microscopic model, despite its concreteness and the use of the state-of-the-art QMC simulation, fails to reproduce the QA effects based on realistic model parameters. Whether or not one considers this warrants publication in this journal may be different from one person to another. I have to say I would not recommend it.

Below is an additional comment for the authors to consider:

1) The interpretation of the SSE MC step needs a clarification about the update algorithm, because the loop or directed-loop updates in general do not correspond to individual spin flips that would take place both in quantum annealing or thermal annealing processes in experiments.

Reviewer #3 (Remarks to the Author):

I thank the authors for responding to my questions and comments. In the latest manuscript, their main theoretical results are now based on the SSE QMC simulation. I would not further comment on their responses about the perturbative MC simulation because the corresponding results have been removed from the manuscript. In any case, I fully appreciate the additional efforts taken by the authors, where they finally decided to take the current standard technique (after many stages of revisions).

With that said, I have to point out that their simulation was merely able to provide a qualitative description of QA effects -- "these simulations unequivocally demonstrate QA effects caused by the small transverse field (...), providing a qualitative description of the previously mentioned experimental observations." The magnitude of the transverse field relevant to their experiment is below 7 mK, whereas the relaxation of the magnetization in this energy scale barely shows any enhancement of the relaxation (Figs. 4f and 4i). Since the relaxation is too slow in simulations and the required magnitude for the transverse field (0.02K?) is at least around 3 times larger (which I think is a rather optimistic estimate) than 7 mK, it seems to me that claiming any connection with their experiments is hard. Although the authors mention to structural defects or internal transverse magnetic fields, these conjectures are no more than speculative. Regarding the internal energy mentioned in the new manuscript (Figs. 4d, 4e, 4g, and 4h), as I mentioned in my previous report, the decrease of energy would be found in any second order perturbative process (that is, uncorrelated spin flips back and forth) and I would not consider it as an enough evidence of QA towards a different ground state.

To conclude, my assessment of the manuscript is such that although the experiments might show some interesting QA effects, the microscopic model, despite its concreteness and the use of the state-of-the-art QMC simulation, fails to reproduce the QA effects based on realistic model parameters. Whether or not one considers this warrants publication in this journal may be different from one person to another. I have to say I would not recommend it.

Response: We appreciate the feedback from Reviewer #3. However, we wish to reiterate that our heat transport measurements, which also exhibit clear quantum annealing effects around 1 K, were conducted at a transverse field of up to 0.07 K, significantly smaller than the critical transverse field of ~ 28 K, and more than three times larger than 0.02 K (please refer to Fig. 5 in the main text). The decrease in energy, as illustrated in Figs. 4d and 4e, has been thoroughly discussed in the fourth paragraph of the "Quantum Monte Carlo simulations" section and cannot be attributed solely to the equilibrium-state energy decrease. Furthermore, the increase in longitudinal magnetization induced by the transverse field (see Fig. 4f) clearly contradicts the equilibrium-state magnetization decrease. Hence, we maintain our conclusion that the SSE-QMC simulations unequivocally demonstrate quantum annealing effects at ~ 1 K in a transverse field down to ~ 0.02 K.

Actions taken: We have referenced our heat transport measurements at the end of the fourth paragraph in the "Quantum Monte Carlo simulations" section.

Below is an additional comment for the authors to consider:

1) The interpretation of the SSE MC step needs a clarification about the update algorithm, because the loop or directed-loop updates in general do not correspond to individual spin flips that would take place both in quantum annealing or thermal annealing processes in experiments.

Actions taken: Following the suggestions of Reviewer #3, we have included a clarification about the SSE-QMC update algorithm in the "Methods" section.